# A machine learning approach to integrate big data for precision medicine in acute myeloid leukemia

Su-In Lee [1,2,3], Safiye Celik[1], Benjamin A. Logsdon [4], Scott M. Lundberg[1], Timothy J. Martins [5], Vivian G. Oehler[6,7], Elihu H. Estey[6,7], Chris P. Miller[7], Sylvia Chien[7], Jin Dai[7], Akanksha Saxena[7], C. Anthony Blau[3,7] & Pamela S. Becker [3,6,7]

Cancers that appear pathologically similar often respond differently to the same drug regimens. Methods to better match patients to drugs are in high demand. We demonstrate a promising approach to identify robust molecular markers for targeted treatment of acute myeloid leukemia (AML) by introducing: data from 30 AML patients including genome-wide gene expression profiles and in vitro sensitivity to 160 chemotherapy drugs, a computational method to identify reliable gene expression markers for drug sensitivity by incorporating multi-omic prior information relevant to each gene's potential to drive cancer. We show that our method outperforms several state-of-the-art approaches in identifying molecular markers replicated in validation data and predicting drug sensitivity accurately. Finally, we identify SMARCA4 as a marker and driver of sensitivity to topoisomerase II inhibitors, mitoxantrone, and etoposide, in AML by showing that cell lines transduced to have high SMARCA4 expression reveal dramatically increased sensitivity to these agents.

[1] Paul G. Allen School of Computer Science and Engineering, University of Washington, 185 E Stevens Way NE, Seattle, WA 98195, USA. [2] Department of Genome Sciences, University of Washington, 3720 15th Ave NE, Seattle, WA 98195, USA. [3] Center for Cancer Innovation, University of Washington, 850 Republican Street, Seattle, WA 98109, USA. [4] Sage Bionetworks, 1100 Fairview Ave N, Seattle, WA 98109, USA. [5] Quellos High Throughput Screening Core, University of Washington, 850 Republican Street, Seattle, WA 98109, USA. [6] Clinical Research Division, Fred Hutchinson Cancer Research Center, 1100 Fairview Ave N, Seattle, WA 98109, USA. [7] Division of Hematology, Department of Medicine and Institute for Stem Cell and Regenerative Medicine, University of Washington, 850 Republican Street, Seattle, WA 98109, USA. Su-In Lee and Safiye Celik contributed equally to this work. Correspondence and requests for materials should be addressed to S.-I.L. (email: suinlee@cs.washington.edu)

The research and development process for new drugs remains challenging and expensive. Nonetheless, the repertoire of potential cancer drugs continues to expand, with more than 1200 cancer medicines in clinical development in the U.S.[1]. However, cancers that appear pathologically similar often respond differently to the same drug regimens. Thus, methods to better match patients to the existing chemotherapy drugs are in high demand.

The growing availability of genome-wide expression data and in vitro drug sensitivity data from cancer cell lines has enabled a data-driven approach to identifying molecular markers by finding robust statistical associations between genes and drugs. The Cancer Genome Project (CGP) tested 130 drugs in 639 cell lines, with a mean of 368 cell lines tested for each drug[2]. The Cancer Cell Line Encyclopedia (CCLE) tested 479 cell lines for sensitivity against a panel of 24 drugs[3]. These studies employed a penalized (elastic net) regression method[4] to identify novel associations between gene expression levels and drug sensitivity measures. While both CGP and CCLE evaluated large numbers of cell lines, some of the most interesting associations were detected by focusing analyzes within, rather than across, tumor types. Consistent with this, a study by Heiser et al.[5] was able to identify novel associations using a much smaller panel of 49 breast cancer cell lines with sensitivity to a panel of 77 compounds.

This paper presents in vitro drug response profiles for 160 chemotherapy drugs along with genome-wide gene expression from 30 patients with acute myeloid leukemia (AML) (Supplementary Data 1). For AML, publicly available data from CGP and CCLE include only 14 cell lines. Conventionally, one tests for associations between gene expression levels and drug sensitivity measures by: (1) measuring pairwise association between each gene and each drug, or (2) performing a penalized regression for each drug using all genes as potential molecular markers, as was done in the CCLE and CGP drug sensitivity studies (Fig. 1a). However, drug response could be associated with gene expressions that do not reflect the underlying drug's biological mechanism (i.e., false positive associations), and therefore, results often do not replicate in another data set[6]. This discrepancy can happen due to biological confounders (disease subtypes or heterogeneity), experimental confounders (sample ascertainment), or technical confounders (e.g., batch effects). Previous studies also raised concerns with respect to drug sensitivity assay robustness[7]. The high-dimensionality of data (i.e., when the number of gene-drug pairs greatly exceeds the number of samples) increases the multiple hypothesis testing burden and the chance of false positive gene-drug associations.

Successful attempts to reduce false positives by incorporating prior information have occurred in genome-wide association studies. Li et al.[8] proposed a prioritized subset analysis: they pre-selected a prioritized subset of single-nucleotide polymorphisms (SNPs) from candidate genes or regions and applied false discovery rate (FDR) correction within this subset to make it more likely that these SNPs would be selected. Roeder et al.[9] and Genovese et al.[10] up- or down- weighted the association p-value for each SNP, and the resulting weighted p-values were subsequently used in FDR corrections; the p-values of SNPs with significant associations in prior linkage analyzes were lowered. Although these methods increase the power to detect causal SNPs in genome-wide association studies, they do not apply when it is not obvious how to define candidate genes or prioritize a subset of genes, especially when multiple sources of prior information are available.

One simple way to use prioritized subset analysis[8] considers the genes frequently mutated in AML as candidate genes. However, relying on only mutation information is unlikely to prove successful. Many cancer mutations are passengers (i.e., not

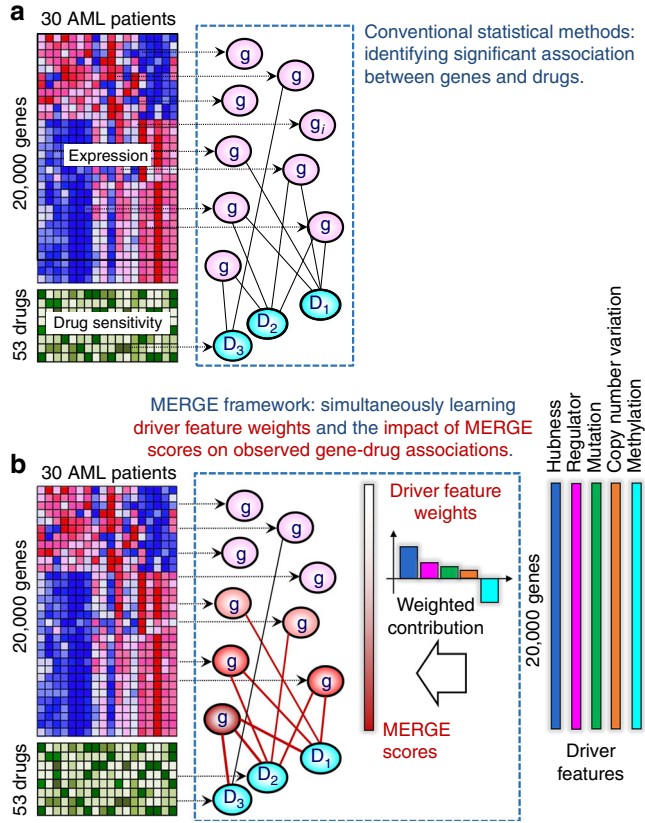

**Fig. 1** Conventional statistical methods vs. MERGE. **a** Conventional methods identify gene expression markers for drugs based on expression data and drug sensitivity data. They measure the statistical significance of associations between expression levels for each gene and sensitivity measures for each drug. **b** The MERGE framework models the marker potential (MERGE score) of each gene based on a weighted combination of the gene's driver features. MERGE simultaneously learns the driver feature weights (and correspondingly, MERGE scores for all genes) and the impact of the MERGE score on the observed gene-drug associations

drivers), and some genes' driver roles may not be reflected in mutation data but in other data types, such as epigenomic, copy number variation (CNV) and even gene expression data. For example, our prior study[11] suggested that hub genes in a gene network inferred purely based on gene expression data might indicate tumor driver events. To reliably identify gene expression markers for drug response, we must precisely determine how much each piece of relevant prior information contributes to the gene's marker potential.

We present MERGE (mutation, expression hubs, known regulators, genomic CNV, and methylation), a novel computational method that identifies reliable gene expression markers using a principled way of integrating multi-omic prior information relevant to disease processes (Fig. 1b). MERGE learns from data how much each of the following driver features contributes to genes' potentials to drive cancer progression: (1) mutations based on the AML study from The Cancer Genome Atlas (TCGA)[12], (2) hubness in a gene expression network inferred from publicly available expression data sets, (3) the gene's known regulatory role based on gene annotation databases[13], (4) genomic CNV status based on the AML study from TCGA, and (5) methylation status based on the AML study from TCGA. We model each gene's marker potential (i.e., the prior probability that the gene is a reliable molecular marker for drugs) as a weighted combination of the gene's driver features. Through an iterative procedure, the

MERGE algorithm jointly learns these weights and the degree of impact that genes' marker potentials have on observed gene-drug associations (Fig. 1b). We show that MERGE outperforms several state-of-the-art approaches in identifying molecular markers replicated in validation data and predicting drug sensitivity accurately. We experimentally validate *SMARCA4* as a molecular marker and driver of sensitivity to topoisomerase II inhibitors, mitoxantrone and etoposide, in AML by showing that cell lines transduced to have high *SMARCA4* expression show dramatically increased sensitivity to these agents.

## Results

**Data collected from 30 AML patients**. We measured genome-wide gene expression (Supplementary Note 1) and in vitro drug sensitivity (Methods section) to a panel of 160 chemotherapy drugs and targeted inhibitors across 30 AML patient samples (Supplementary Data 1). The customized drug panel we used contained 62 drugs approved by the U.S. Food and Drug Administration (FDA) and encompassed a broad range of drug action mechanisms (Supplementary Data 2). The other drugs, investigational agents, have been studied in cancer patients. We chose 53 drugs that exhibited activity (cell viability ≤50%) against at least half of the patient samples (Supplementary Table 1). As was done previously in the CCLE study[3], we processed the drug sensitivity data by curve fitting and then extracting summary statistics. We used the area under the curve (AUC) throughout the paper because it represents an average of drug sensitivity across a range of drug concentrations; indeed, AUC showed by far the strongest association with gene expression levels (Supplementary Note 2). Supplementary Data 3 describes usual evaluation (including risk group category and cytogenetic features), response to treatment, and duration of remission. Supplementary Note 3 summarizes the clinical information and describes our analysis on the consistency between clinical data and our in vitro drug sensitivity data. In brief, we showed a statistically significant association between *FLT3* mutation status and 12 drugs known to have a *FLT3* inhibitory role. Statistical significance of the association between the complete remission (CR) status and the AUC across all 53 drugs.

**Drug sensitivity assayed in 14 AML cell lines**. We utilized cell lines to intensify our focus on the hypotheses for which we could provide additional experimental evidence, since it is easier to perform overexpression or knockout experiments on cell lines than on primary patient tissues. However, we noted a very small overlap between our 160 drugs and the drugs tested on the 14 AML cell lines in the CCLE data (two drugs overlapping, each tested on three AML cell lines). Thus, for effective computational and experimental validation of the significant gene-drug associations discovered in the patient data, we measured in vitro drug sensitivity of 14 AML cell lines to the same set of 160 drugs in our high-throughput assay (Supplementary Data 1) while we used publicly available expression data of the 14 AML cell lines from CCLE. We observed a statistically significant overlap (Fisher's exact test $p$-value $= 3 \times 10^{-6}$) of gene-drug pairs with significant association $p$-values between our discovery data from 30 AML patient samples and the CCLE validation data. Our unique validation setting let us measure the testability of discovered associations. We also surmised that in vitro drug sensitivity data that we measured on the AML cell lines, besides on the AML patient samples, would provide a valuable resource to the broader research community to generate or test hypotheses relevant to personalized medicine in AML.

**The MERGE algorithm provides a new way to prioritize genes**. Our MERGE algorithm provides a new way to prioritize gene-drug associations by incorporating prior information on genes' relevance to AML and potential to drive it (Methods section). MERGE learns a priority score for each gene, called a MERGE score, based on the gene's driver features: (1) mutation, (2) expression hubness, (3) whether the gene has a known Regulatory role, (4) genomic CNV, and (5) methylation. These driver features were extracted from publicly available sources, such as TCGA AML study[12], AML expression studies[14], and gene annotation databases. The details on the sources and the pre-processing of the driver features are included in Supplementary Notes 4–6.

The MERGE score represents a prior probability that the gene is a reliable (i.e., likely driven by biological mechanisms, not confounders) molecular marker for response to drugs, modeled as a weighted combination of the gene's MERGE features (Fig. 1b). The MERGE algorithm jointly learns these driver feature weights and how the MERGE score of genes explains the observed gene-drug associations. Genes with high MERGE scores (i.e., high marker potentials) tended to have many observed associations with drugs. The learned driver feature weights provide new insights into what kind of molecular data is most informative of a gene's potential to be a reliable marker for therapeutic response. After the MERGE score of each gene is estimated, we considered the top $N$ genes based on MERGE scores as a prioritized subset and then selected the gene-drug pairs with significant association $p$-values (genome-wide FDR corrected $p$-value <0.1) (Supplementary Note 7).

**Expression hubness significantly determines the MERGE scores**. The driver feature weights learned by the MERGE algorithm (Methods section) indicate the relative importance of each driver feature on the MERGE scores (Fig. 1b). As described in Supplementary Note 8, the MERGE score of gene $i$ is defined as $\left( \sum_{k=1}^{5} v_k d_{ik} \right)$, where $v_k$ is the $k$th driver feature weight and $d_{ik}$ is the $k$th driver feature of gene $i$. Expression hubness (i.e., number of neighbors in a gene network estimated based on publicly available AML expression data) has the highest weight (Fig. 2a) and makes the largest contribution to the MERGE scores (Fig. 2b, Supplementary Data 4).

This implies that gene expression data provide more information about a gene's potential to predict drug response than other types of data, perhaps for the following reasons: (1) Gene expression data can reflect downstream effects of genetic or epigenetic changes that may not have been detected by existing mutation, CNV and methylation profiles. (2) Expression hubness was estimated from a larger number of patients because expression data are the most common type of molecular data from disease studies. (3) Expression hubness has been considered likely to indicate selective pressure in tumor genome evolution and to drive events[11].

Identifying expression hubs has therefore been considered a powerful complementary way to identify candidate tumor drivers that are hard to detect from sequence data due to a large number of passenger mutations. The importance of the expression hubness feature (Fig. 2) suggests that these candidate tumor drivers might prove promising markers for drug response. The methylation feature is negatively weighted, consistent with prior knowledge that methylation in a promoter region silences the corresponding gene[15].

**Overview of our statistical and biological findings**. We evaluated our MERGE algorithm in four different ways and we observed that the results were very promising in all four categories of evaluation we performed. Following is a summary of

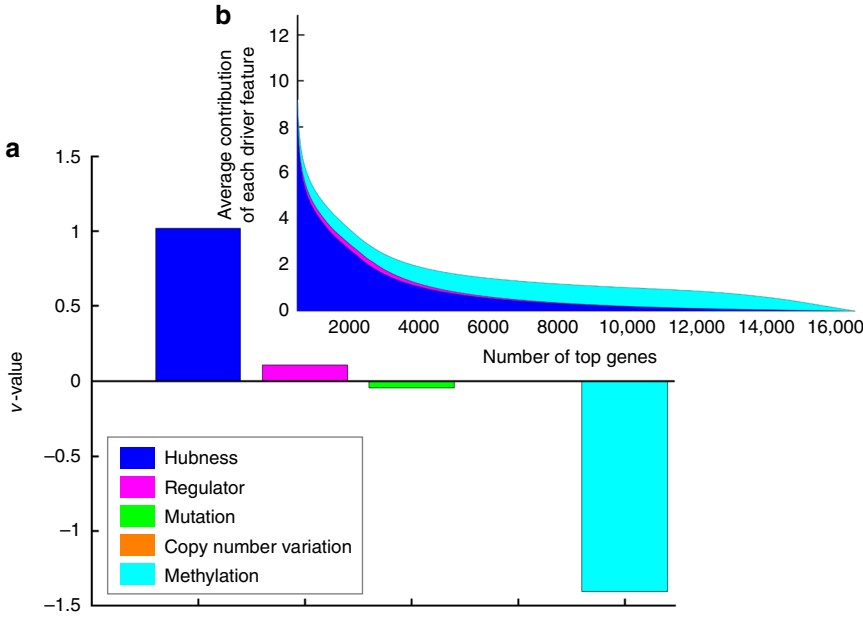

**Fig. 2** Importance of each driver feature in predicting the drug response based on the MERGE algorithm. **a** Learned driver feature weight values. The methylation feature has a negative weight, consistent with our prior knowledge that when DNA is methylated in the promoter region, the corresponding genes are inactivated and silenced. **b** We sort genes based on the MERGE score (x-axis) and plot the sum of the contribution of all driver features to the MERGE score (i.e., weighted combination of driver features) (y-axis). We decomposed the weighted combination into the five driver features and indicated the magnitude of the contribution of each feature (driver feature weight × driver feature value) with different colors. Expression hubness contributes the most to the score, followed by regulatory function and (lack of) methylation

those categories: (1) Consistency of significant gene-drug associations in left-out test data. We compared MERGE to four other methods in terms of consistency of significant gene-drug associations, where we used all 30 patient samples for discovery and validated the discovered associations using (i) 14 cell lines, (ii) independent patient data from 12 new AML specimens. (2) Consistency of significant gene-drug associations within drug functional classes. If a gene $X$ were associated with a drug $Y$, $X$ would also be likely associated with another drug $Y'$ with the same mechanism of action (e.g., sunitinib and tandutinib are in the 'Flt3 inhibitor' class) (Supplementary Data 2). We compared MERGE to four other methods in terms of within-drug class consistency—the extent to which the gene-drug association was conserved across drugs in the same functional category. (3) Prediction of patient drug response. We compared MERGE to three other methods for evaluating consistency of patient rankings based on actual vs. predicted drug sensitivity. We tested in two ways: (ii) We used one batch containing 12 patient samples for training and a different batch containing 12 patient samples for validation (and vice versa). (ii) We used a leave-one-out cross validation (LOOCV) test to obtain the predicted drug sensitivity across 30 patient samples. (4) Biological interpretation and experimental validation. We discussed top-ranked genes for several drug classes based on our MERGE prioritization method and their associations with the corresponding drugs. Finally, we described results of the experimental validation on one of the top-ranked genes, *SMARCA4*, whose expression is significantly associated with increased sensitivity to mitoxantrone and etoposide.

**Consistency of gene-drug associations in left-out data**. We compared MERGE to four conventional methods: (1) Pearson's *P*-value of correlation. For each gene-drug pair, we computed the *t* statistics and the associated *p*-value, measuring the correlation in a univariate linear regression model. We then selected the

gene-drug pairs with the smallest association *p*-values. (2) Spearman *P*-value of correlation. For each gene-drug pair, we measured the rank association using the Spearman correlation and selected the gene-drug pairs with the smallest association *p*-values. (3) ElasticNet. For each drug, we solved the elastic net optimization problem[4] using gene expression as input, as done by Barretina et al.[3] and Garnett et al.[2], and selected the gene-drug associations with the strongest weight. (4) Multi-task learning. We used the multi-task learning method[16] implemented by Pong et al.[17], which considers each drug as a different task. We then selected the gene-drug associations with the strongest weight.

We discovered gene-drug associations within the data from all 30 samples, and tested them on two independent data sets: (a) 14 CCLE cell lines (Fig. 3a, b) 12 additional AML patients who had relapsed or were refractory to at least two (up to six) prior regimens (Fig. 3b). Supplementary Note 9 shows the clinical information on the additional 12 patients, and Supplementary Data 5 presents the gene expression (processed RNA-seq) and the drug sensitivity (AUC) data from the same patients.

Each method prioritized gene-drug pairs differently (Supplementary Note 10). For evaluation, we computed the true discovery rate (that is, how many significant associations were replicated in the left-out test data) when considering the top $N$ genes per drug on average (i.e., $53 \times N$ gene-drug pairs in total). Specifically, we computed the consistency rate (y-axis)—defined as the number of significant gene-drug associations replicated in the left-out test data divided by the total number of significant gene-drug associations within the selected $53 \times N$ gene-drug pairs—for varying values of $N$ (x-axis) from one to all genes. In both settings (Fig. 3a, b), MERGE showed a much higher consistency rate for high-scoring genes (small $N$ values) than the other two methods and the random ordering of genes (gray lines). As $N$ increased, methods had more similar consistency rates because their top $N$ genes became more similar.

Unlike the initial 30 samples, the additional 12 samples were highly refractory, and many of them exhibited extremely poor

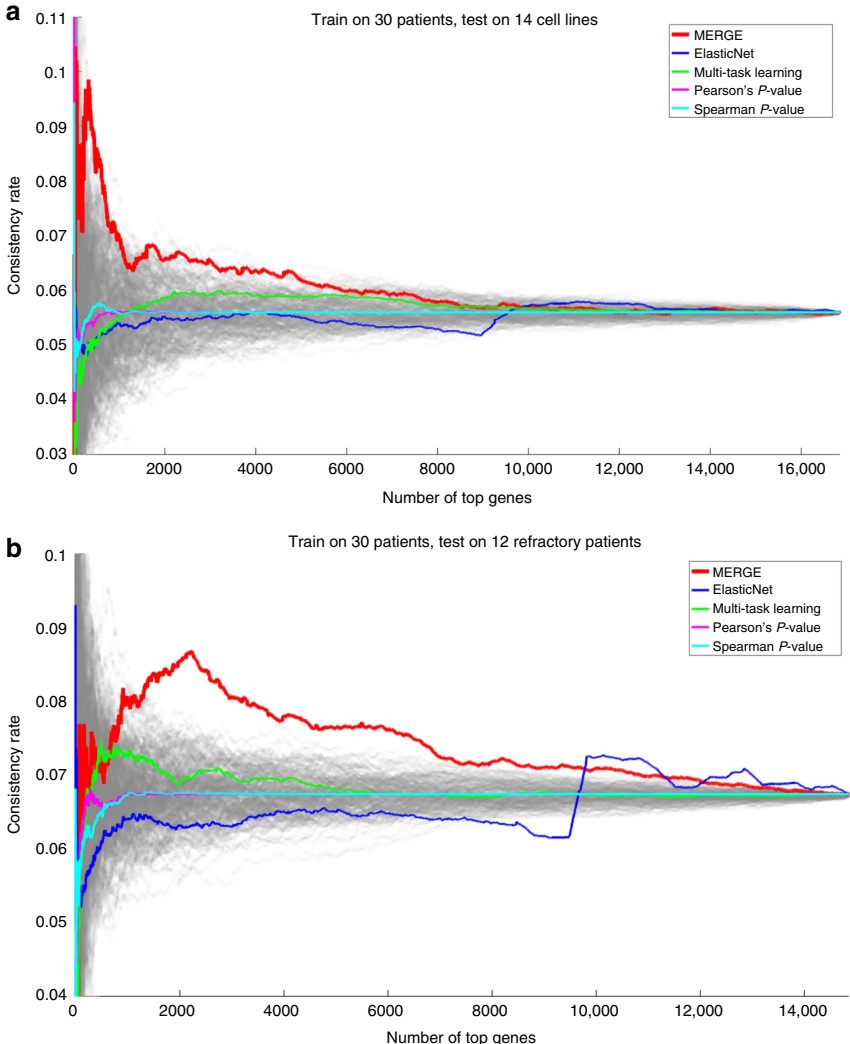

**Fig. 3** Comparison of MERGE with four alternative methods in terms of the percentage of the significant associations replicated in the left-out test data. We discovered gene-drug associations within the data from all 30 samples, and we tested on **a** the 14 cell line samples, and **b** the data from the additional 12 refractory patient samples. Each gray line corresponds to a random ordering of all genes. We note that the x-axis in **b** contains a lower total number of genes; some of the genes existing in the microarray gene expression data from the 30 patient and cell line samples did not exist in the RNA-seq data from the 12 refractory patient samples

risk features. This could potentially account for the lower consistency rate in Fig. 3b even though both the discovery and validation samples were obtained from patient specimens. We did not expect that highly refractory patients (refractory after 2–6 prior regimens) would exhibit the same drug sensitivity patterns as the 30 patients with newly diagnosed disease or first or second relapses. In addition, we surmised that the refractory patients may have activated other survival pathways and mechanisms of drug resistance. Still, even in this challenging validation setting, MERGE outperformed four alternative methods.

Three reasons accounted for the poor performance of the ElasticNet and Multi-task learning methods in Fig. 3a, b. First, these multiple regression methods, intended to solve a prediction problem, were not specifically designed to capture robust gene-drug associations. Conversely, MERGE was designed to aggressively decrease the number of false positive gene-drug associations by incorporating prior knowledge about the genes' potential to drive the disease. Second, since this problem was ultra-high-dimensional (30 samples and ~17 K variables), multiple regression methods were likely to learn models too complex to identify robust gene-drug associations, even with regularization (e.g.,

elastic net penalty). Finally, since multiple regression methods model each response as an aggregated effect of multiple features, highly correlated features would share a fixed amount of weight. This would assign a small magnitude of weight to each of many correlated features. Many robust gene-drug pairs whose associations would have been replicated in validation data were likely to have been eliminated in this way.

MERGE uses a strong prior and this might have been an important factor for MERGE's strong feature consistency result. To show that the training data are also a critical factor for the high consistency rate of MERGE, we performed MERGE on 100 different permutations of the data where the training samples in the drug response data are shuffled. We then tested the learned models on both 14 cell lines and the 12 refractory samples, similarly to Fig. 3. As shown in Supplementary Fig. 1, the MERGE run using the original sample ordering achieved a higher consistency rate than most of the permutation runs, and many of the permutation runs perform worse than the competing methods, which shows that MERGE makes use of the information in the training data and the prior information alone is not very helpful when the training data is random.

**Drugs in the same classes exhibited similar response pattern.** The 53 drugs we considered were classified into 24 broad classes based on their mechanism of action; 15 classes contained >1 drug (Supplementary Data 2). Sixteen of the 53 drugs were shared across two classes, while the other 37 drugs were in a single class. It was easier to define the class of older, more classic drugs. It became more difficult for inhibitors that had differential actions for different targets, so we kept them in a more general class. Different drugs with similar mechanisms of action were expected to show similar response across patients and show similar patterns of gene-expression association. To test this hypothesis in an unbiased way, we first applied an agglomerative hierarchical clustering approach to compare the AUC values of different drugs across 30 patient samples. Figure 4a shows that drugs in at least 10 of 15 classes (with >1 drug) were clustered, which indicates that drugs with the same mechanism of action tended to have similar response patterns (Supplementary Note 11).

For each gene, we also examined whether the drugs with which the gene was significantly associated tended to be clustered into the drug mechanism classes. We computed each gene's specificity measure, which we referred to as its drug class specificity (DCS) score (Supplementary Note 12). For each gene significantly associated with at least one drug, we then estimated the significance of its DCS score by comparing it with 1000 random DCS scores for the same gene, each from a permutation test where we shuffled the class labels of all drugs. Figure 4b shows a

QQ plot of the empirical DCS score $p$-values from 1000 random permutation tests ($y$-axis) against the Uniform(0,1) distribution quantiles ($x$-axis) for each gene. The empirical DCS $p$-values were significantly lower than random (permutation test $p$-value: 0.029), indicating the statistical significance of the DCS in gene expression dependency of the drugs overall.

**Specificity of gene-drug associations to drug classes.** If a gene's expression level were specifically associated with drugs in the same mechanism class, we would have higher confidence in the observed associations resulting from the underlying biological mechanisms. Genes with expression levels associated with too many drugs in a diverse set of classes would be less likely to be true markers, and the observed associations would more likely be due to confounders. Here, we used the specificity of gene-drug associations to a drug mechanism class as an evaluation metric, and we used the DCS score as the evaluation metric for each gene (as described above). We took the average DCS score ($y$-axis) over the genes associated with the top gene-drug pairs (i.e., $53 \times N$ gene-drug pairs in total) for varying $N$ values ($x$-axis) (Fig. 4c). MERGE showed a much higher degree of drug mechanism class specificity than the alternative methods.

**Drug sensitivity prediction performance.** We next compared MERGE to the following methods that predict patient drug

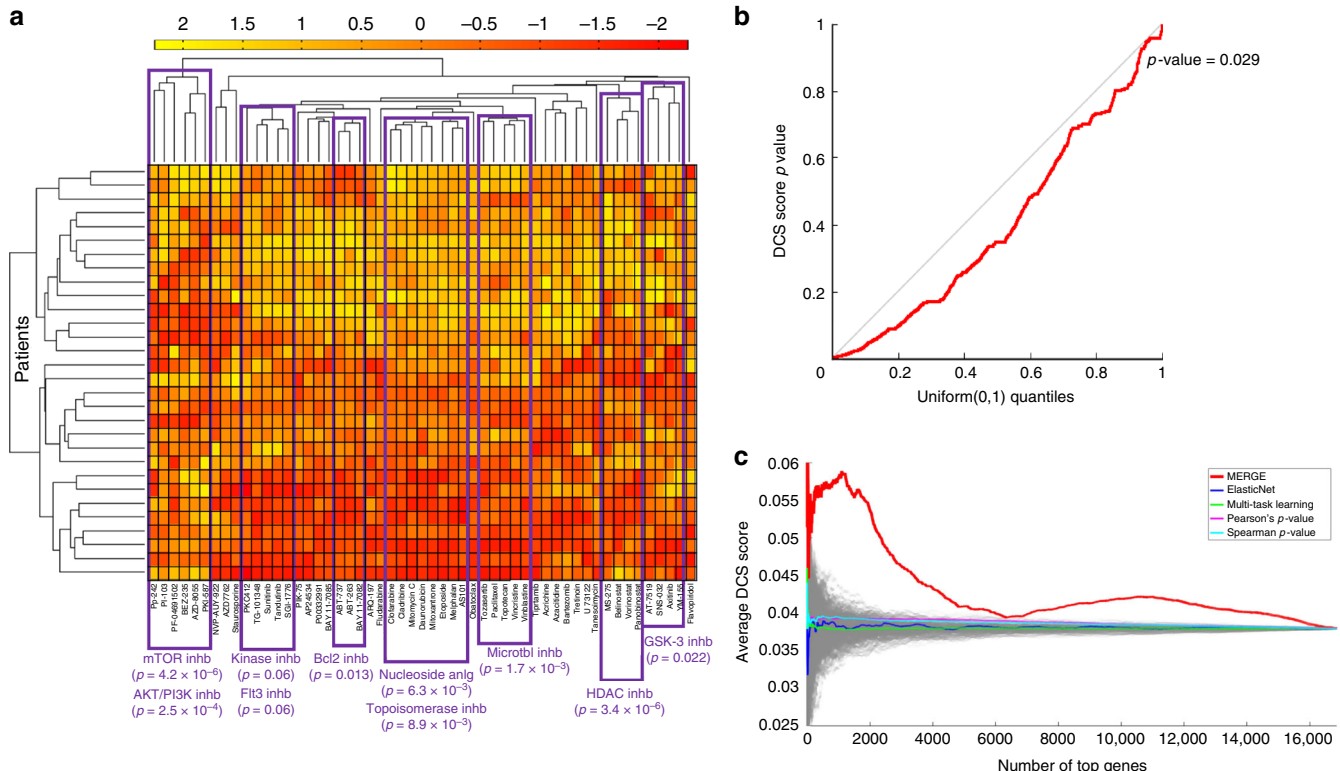

**Fig. 4** Drug class specificity (DCS) of the associations prioritized based on the MERGE algorithm. **a** The result of the agglomerative hierarchical clustering that is applied to the AUC values of all drugs across 30 patient samples. AUC values are standardized before applying the hierarchical clustering, and the color bar at the top represents the standardized AUC values. For several branches in the resulting dendrogram, we report the drug classes that have significant enrichment with the drugs in that branch; we also report the corresponding Fisher's exact test $p$-values. The result shows that drugs in at least 10 of 15 classes (that contain >1 drug) are expectedly grouped together in the dendrogram; for each group, Fisher's exact test $p$-value associated with the overlap between the drug class and the dendrogram group is $\leq 0.06$. **b** A QQ (quantile–quantile) plot of the observed DCS score $p$-values from 1000 random permutation tests for all genes with at least one significant gene-drug association. In each permutation test, drug labels were shuffled. Dots falling below the diagonal imply that the overlap with known drug classes is more significant than would be expected by random chance (permutation test $p$-value = 0.029). **c** For varying $N$ ($x$-axis), we plotted the average DCS score over the $N$ genes ($y$-axis) associated with the top ($53 \times N$) gene-drug pairs based on the five methods in comparison

response: (1) ElasticNet[4], (2) multi-task learning[16,17], and (3) Bayesian multi-task multiple kernel learning (MKL), the winner of the NCI-DREAM Drug Sensitivity Prediction Challenge[18]. Like MKL, we compared these methods in terms of the consistency among the ranking of patients based on their actual versus predicted drug sensitivity.

We considered the following two settings: (a) we trained a prediction model based on 12 samples in one batch and tested on the 12 samples in another batch, and vice versa; then we averaged prediction accuracy results (Fig. 5a, b). We measured the prediction performance via LOOCV using all 30 patient samples (Fig. 5b).

In Fig. 5a, b, we compared MERGE (*y*-axis) to the three other methods (*x*-axis) in terms of prediction performance measured by rank correlation between predicted and actual drug response in the test set across patients. Each dot corresponds to one of the 53 drugs, and each color to one of the methods compared to MERGE. In both experimental settings (a and b), MERGE performed competitively with the alternative methods in terms of prediction performance averaged over all drugs. We performed a one-sided Wilcoxon signed-rank test to show the significance of the outperformance of MERGE relative to the methods in comparison. In fact, the *p*-values are significant at a $p \leq 0.007$ level for four out of six comparisons and at a $p \leq 0.1$ level for the other two. We show the *p*-values in the legend of Fig. 5a, b for each comparison, and Supplementary Fig. 2a, b show the detailed performance for each of the drugs. As shown in the legend of Supplementary Fig. 2a, b, MERGE achieves the best prediction performance for a higher number of drugs than each of the three alternative methods. Indeed, in the LOOCV test (Supplementary Fig. 2b), MERGE achieved the best prediction performance for 62% of the drugs.

**MERGE identifies the roles of several drug response markers**. We now interpret the gene-drug associations identified by MERGE, which offer the potential to make novel discoveries about molecular markers (Fig. 6). We seek to derive hypotheses likely to lead to discoveries or experimental validation targets. Therefore, we further narrowed significant gene-drug associations to: (1) those that were consistently significant in cell lines so we could perform experimental validations, and (2) those that showed a high degree of specificity for a drug mechanism class. For each of the 24 drug mechanism classes, we considered the high MERGE-scoring genes whose associations with drugs were specific to that class (Fisher's exact test *p*-value <0.05) and whose associations to that class were conserved in cell lines. Supplementary Table 2 lists these genes for the 20 drug classes that had at least one class-specific gene. Supplementary Data 6 shows the entire list of the genes for each class and the corresponding results in detail. Figure 6a depicts a heat map that shows the level of specificity of each gene (row) to each drug class (column), measured by $-\log_{10}$ [Fisher's exact test *p*-value] for the top three MERGE-scoring genes in each drug class. Figure 6b highlights the drugs associated with each gene. In Fig. 6b, the red color indicates a negative association between gene expression and drug AUC measure (i.e., high expression indicates low AUC and hence sensitivity), while green indicates a positive association (i.e., high expression indicates resistance). We show in Supplementary Fig. 3 the heat maps for the four alternative methods with which we compared MERGE (in Figs 2, 3c). Supplementary Fig. 4 shows the amount of contribution of each driver feature on the MERGE score for the genes shown in Fig. 6, b.

The following sections summarize the eight top-ranked genes in some of the major drug classes (*FLT3, CASP8AP2, L2HGDH, MNT, BAZ2B, MZF1, BEX2,* and *SMARCA4*) that are highly likely to have notable biological significance in leukemia. We

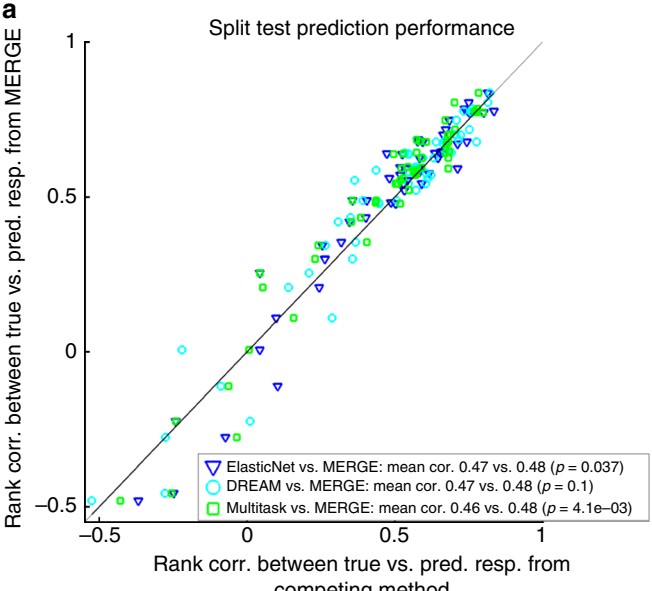

**a**

Split test prediction performance

ElasticNet vs. MERGE: mean cor. 0.47 vs. 0.48 (*p* = 0.037)
DREAM vs. MERGE: mean cor. 0.47 vs. 0.48 (*p* = 0.1)
Multitask vs. MERGE: mean cor. 0.46 vs. 0.48 (*p* = 4.1e−03)

Rank corr. between true vs. pred. resp. from MERGE (y-axis)
Rank corr. between true vs. pred. resp. from competing method (x-axis)

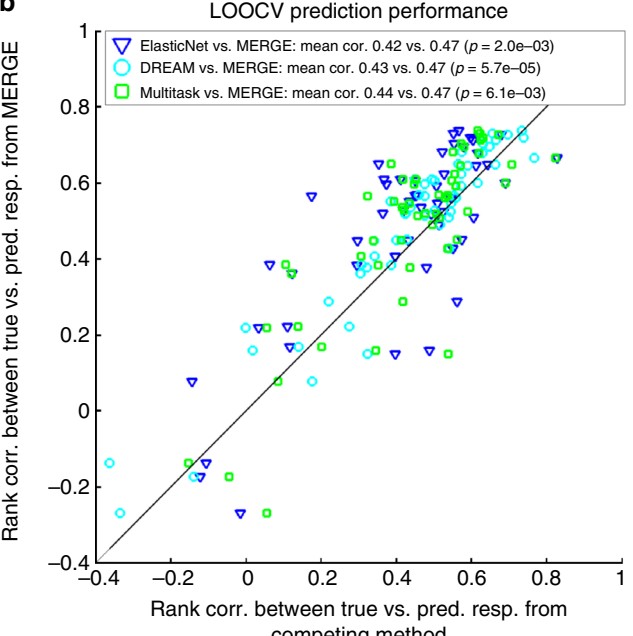

**b**

LOOCV prediction performance

ElasticNet vs. MERGE: mean cor. 0.42 vs. 0.47 (*p* = 2.0e−03)
DREAM vs. MERGE: mean cor. 0.43 vs. 0.47 (*p* = 5.7e−05)
Multitask vs. MERGE: mean cor. 0.44 vs. 0.47 (*p* = 6.1e−03)

Rank corr. between true vs. pred. resp. from MERGE (y-axis)
Rank corr. between true vs. pred. resp. from competing method (x-axis)

**Fig. 5** Comparison of prediction performance of MERGE to the prediction performances of three other methods: ElasticNet, Bayesian multi-task MKL and multi-task learning. **a** One batch of the samples is used for training, and a different batch is used for validation. **b** LOOCV setting. Performance is measured in terms of the Spearman correlation of the predicted response with the actual response. This evaluation metric was used in the NCI-DREAM Drug Sensitivity Prediction Challenge. Each dot corresponds to a different drug, and each color to a different method's prediction on the *x*-axis compared to the MERGE prediction on the *y*-axis. The mean correlation from each of the methods in comparison and the associated *p*-values from a one-sided Wilcoxon signed-rank test are reported in the legend

observed that for five of these eight genes (*SMARCA4, FLT3, MNT, BAZ2B,* and *MZF1*), MERGE provided a unique path toward identifying their roles that the other four methods simply could not identify (Supplementary Table 3). Except for ElasticNet, the alternative methods identified only one of these eight genes, *L2HGDH*.

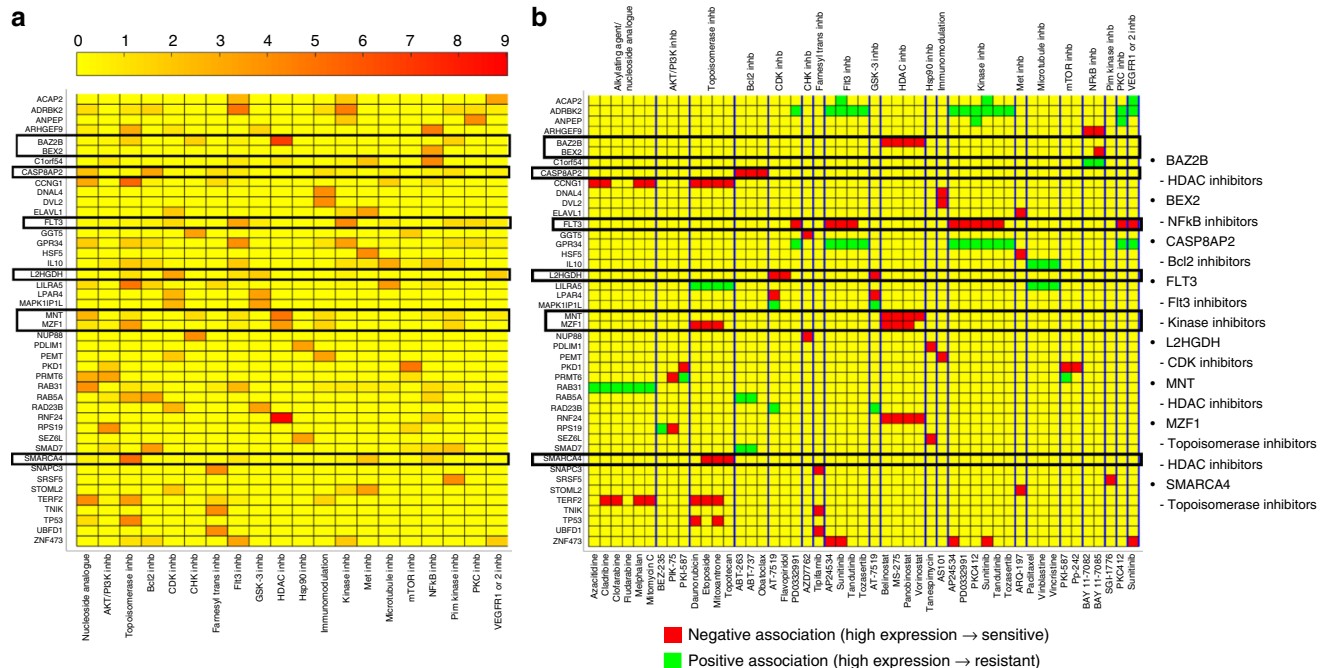

**Fig. 6** The 44 genes in total each of which was identified, by the MERGE approach, as being one of the top three important marker genes for a drug mechanism class. **a** A heat map that shows the level of specificity of each of the 44 genes (row) to each drug class (column) measured by −log$_{10}$ (Fisher's exact test $p$-value). For clarity, we considered only Fisher's exact test $p$-value <0.05 to be significant; other values are indicated in yellow. The drug classes that are not assigned by MERGE any genes with associations specific to the class and consistent in the cell line data are not shown. We highlighted the genes whose biological significance, we discussed in the Results section with black-colored boxes. **b** A heat map that shows the gene-drug association for genes and drug classes shown in **a**. Yellow indicates that the corresponding gene-drug pair does not have a statistically significant association (genome-wide FDR corrected $p$-values <0.1), while green indicates a positive and red a negative association. The drugs are grouped by blue lines based on their classes, and the class names for each group are written on top of the heat map. Drugs that are members of more than one drug class (e.g., sunitinib) are shown multiple times for each class to which the drug belongs. The list on the right shows the genes whose biological significance we discussed in the Results section, and the drug classes they are specific to

**FLT3 expression predicts response to Flt3 inhibitors.** *FLT3* (FMS like tyrosine kinase 3), the top-ranked gene in the Flt3 inhibitor class (Supplementary Table 2), is a significant expression hub and regulator that is highly mutated (Supplementary Fig. 4). *FLT3* mutations are associated with a poor prognosis in AML[19,20]. The principal type that occurs in about 25% of AML patients is internal tandem duplication (ITD). After treatment with some Flt3 inhibitors[21], these patients often develop *FLT3* D835 (kinase domain) mutations, which is present in 8% of patients. One study shows a correlation between *FLT3* expression levels and prognosis among patients with wild-type *FLT3*[22].

Our method led to a new finding: the high expression of *FLT3* is associated with increased sensitivity to the drugs sunitinib, ponatinib (previously AP24534), midostaurin (PKC412), and tandutinib (Fig. 6a, b). Sunitinib is a multi-targeted receptor tyrosine kinase inhibitor FDA-approved for renal cell carcinoma and imatinib-resistant gastrointestinal stromal tumor. Ponatinib, another multi-targeted receptor tyrosine kinase inhibitor, is FDA-approved for chronic myelogenous leukemia that has failed to respond to first-line inhibitors. Tandutinib is an inhibitor of the type III receptor tyrosine kinases *FLT3*, platelet-derived growth factor receptor (PDGFR), and KIT. Midostaurin was studied in a prospective, randomized trial in newly diagnosed patients with *FLT3* mutations undergoing induction, consolidation, and maintenance therapy, and the group that received midostaurin exhibited prolonged event free ($p = 0.0044$) and overall survival ($p = 0.007$) as compared to placebo[23]. Midostaurin recently received breakthrough therapy designation from the FDA for newly diagnosed *FLT3*-mutated AML.

Because *FLT3* mutation status is an important prognostic indicator in AML with the potential to guide therapy, we compared *FLT3* mRNA expression and *FLT3* mutation status in terms of significance of correlation with response to 53 drugs. For (a) patients and (b) cell lines, Supplementary Fig. 5 shows the significance of association achieved by *FLT3* expression level ($y$-axis) vs. by *FLT3* mutation status ($x$-axis) for each drug. More dots appear above the diagonal in both Supplementary Fig. 5a, b, which implies that mRNA level achieved a more significant association than *FLT3* mutation status for a larger number of drugs (36 vs. 17 in patients; 31 vs. 22 in cell lines). For fair comparison, to generate Supplementary Fig. 5a for both *FLT3* mRNA and mutation status, we used only 27 patients for whom mutation status was known. Further, we used Quentmeier et al. [24] as the source for the *FLT3* mutation status for the cell lines.

**CASP8AP2 expression predicts sensitivity to Bcl2 inhibitors.** *CASP8AP2* (caspase 8 associated protein 2) is the top-ranked gene in the Bcl2 inhibitor class (Supplementary Table 2). Overexpression of *BCL2* lets cancer cells evade apoptosis, the process of programmed cell death. The first Bcl2 inhibitor to gain approval for patients, venetoclax, was approved by the FDA on 11 April 2016 to treat the subset of relapsed patients with chronic lymphocytic leukemia that have the deletion of chromosome 17p, which contains the *TP53* gene. The drug is also undergoing evaluation in early phase AML clinical trials (e.g. NCT02203773, NCT02287233 on ClinicalTrials.gov). We found an association between an increased expression of *CASP8AP2* with an increased

sensitivity to Bcl2 inhibitors (Fig. 6b). Caspases mediate the activation of apoptosis, and low expression of *CASP8AP2* has been associated with a poor prognosis in pediatric acute lymphoblastic leukemia[25] and an increased risk of relapse[26]. Moreover, a single report notes a translocation of the poor risk *MLL* (mixed lineage leukemia) gene to the *CASP8AP2* gene in an AML patient with a t(6;11)(q15;q23) translocation[27]. It is therefore not surprising that expression of a gene in the apoptotic pathways (e.g., a caspase associated protein) would be associated with sensitivity to inhibiting the activity of one of the main antagonists of apoptosis. Restoring apoptosis by inhibiting *BCL2* may enhance cell death by increasing the level of expression of apoptotic pathway proteins, such as *CASP8AP2*.

**L2HGDH expression is the predictor for CDK inhibitors.** Cyclin-dependent kinase (CDK) inhibitors prevent proliferation of cancer cells. *L2HGDH* (L-2-hydroxyglutarate dehydrogenase) is an enzyme that catalyzes conversion of L-2-hydroxyglutarate to alpha ketoglutarate. We found that the increased expression of *L2HGDH*, the only gene associated with the CDK inhibitor class (Supplementary Table 2), was associated with increased sensitivity to CDK inhibitors. Elevated serum 2-hydroxyglutarate (2-HG) levels have been associated with isocitrate dehydrogenase (IDH1 and IDH2) mutations in AML[28,29], as well as clinical outcomes[28]. In addition, elevated 2-HG inhibits the function of *TET2*, resulting in DNA hypermethylation[30]. In addition, histone demethylases are competitively blocked by 2-HG[31]. The elevated expression of L2HGDH would be expected to lower levels of 2-hydroxyglutarate and thus abrogate the deleterious secondary effects of DNA and histone methylation. CDK inhibitor genes were the most frequently downregulated genes in IDH1 mutants[32] with high 2-HG levels, thus linking the association we observed with susceptibility to CDK inhibitors with *L2HGDH*.

**MNT and BAZ2B are expression markers for HDAC inhibitors.** We found a correlation between susceptibility to HDAC inhibitors and increased expression levels of *MNT* and *BAZ2B* (Fig. 6b), top two genes in the HDAC inhibitor class (Supplementary Table 2). *MNT* encodes the protein Mnt, a member of the Myc/Max/Mad network of transcription factors that controls cell proliferation, differentiation, and death. It is known to be critical for normal myeloid differentiation in AML cell lines, and its loss leads to proliferation[33]. Mnt has a Sin3-interaction domain (SID) that lets it interact with mSin3A and mSin3B, which recruit histone deacetylases (HDACs), resulting in transcriptional repression[34]. Therefore, a high expression of *MNT* would be predictive of whether inhibiting HDACs has an effect.

*BAZ2B* (Bromodomain Adjacent to Zinc Finger Domain, 2B), is a bromodomain containing protein. The sole function of bromodomain is to recognize acetyl-lysine on histones and non-histone proteins[35]. Histone deacetylases remove acetyl groups from these sites. This suggests that high expression levels of such genes are associated with chemotherapy drug sensitivity for HDAC inhibitors, bringing a new insight into mechanisms that govern individual patient response to chemotherapy involving regulation of chromatin state. Additionally, a popular class of bromodomain inhibitors has shown potential for treating aggressive leukemia[36].

Histone deacetylases impair myeloid differentiation. This is why HDAC inhibitors are potentially useful in therapy and have long been long investigated to treat myelodysplastic syndrome and AML[37]. Various publications address the efficacy of some HDAC inhibitors in clinical trials in AML, including vorinostat, panobinostat, and romidepsin[38–40].

**BEX2 expression is a potential marker for NFkB inhibitors.** NFkB (Nuclear factor kappa B), a transcription factor, is constitutively active in many cancers and inhibits both apoptosis and drug resistance. BEX2, one of the top genes in the NFkB inhibitor class (Supplementary Table 2), is overexpressed in breast cancer and glioma[41]. In KIT-driven AML, NFkB binds to Sp1 and transactivates KIT[42]. In addition, inhibiting both NFkB and JNK is effective in AML expressing the tumor necrosis factor[43]. Lastly, *BEX2* expression occurs in AML with translocations involving the *MLL* gene[44].

**SMARCA4 and MZF1 are markers for topoisomerase II inhibitors.** *SMARCA4* and *MZF1* are the top two genes in the anthracycline/topoisomerase inhibitor/DNA intercalator class (Supplementary Table 2), and both genes are correlated with sensitivity to the drugs in that class (Fig. 6a). The drugs *SMARCA4* is associated with are mitoxantrone, etoposide, and topotecan (Fig. 6b). Mitoxantrone and etoposide are topoisomerase II inhibitors; topotecan is a topoisomerase I inhibitor. *SMARCA4* (SWI/SNF related, matrix associated, actin dependent regulator of chromatin, subfamily a, member 4) is a component of one of the adenosine triphosphate (ATP)-dependent SWI/SNF-like Brg/Brm-associated factor (BAF) chromatin remodeling complexes. Mutations of *SMARCA4* (BRG1) occur in 10–35% of non-small cell lung carcinoma, 15% of Burkitt's lymphoma, 5–10% of childhood medulloblastoma, and less frequently in other cancers[45]. Further, *SMARCA4* mutations characterize small cell carcinoma of the ovary of the hypercalcemic type[46]. Furthermore, one recent study demonstrated a direct role for *SMARCA4* in facilitating decatenation of DNA by topoisomerase II[47]. In leukemia, the core ATPase subunits are BRG/SMARCA4 and BRM/SMARCA2, where BRG/SMARCA4 is essential for the proliferation of both normal hematopoietic stem cells and leukemia stem cells[48].

*MZF1* is a member of the SCAN-zinc finger family of transcription factors, frequently mutated in many types of cancer[49]. Initially, it was believed to have a role in promoting myeloid malignancy[50], but because knockout animals exhibited proliferation of hematopoietic progenitors, it was also thought to potentially suppress hematopoietic malignancy[51]. *MZF1* localizes in the PML-NBs (promyelocytic leukemia nuclear bodies)[52], which are protein complexes involved in post-translational modification of nuclear proteins and response to DNA damage[53]. It was shown that the topoisomerase II inhibitors etoposide and doxorubicin, known to create double-stranded DNA breaks[54], increased the number of PML-NBs by a fission mechanism, and then the number became regulated by cell cycle checkpoint control[53]. Thus, topoisomerase inhibitors directly affect the PML-NBs that incorporate *MZF1*.

**Experimental validation of the marker role of SMARCA4.** Given the high level of specificity of the association between *SMARCA4* and the topoisomerase inhibitor class and prior knowledge of the relationship between *SMARCA4* and topoisomerase II described above, we experimentally validated the association between *SMARCA4* and the topoisomerase II inhibitors (etoposide and mitoxantrone) by overexpressing *SMARCA4*. The U937 leukemia cell line expressed high levels of SMARCA4 protein, while the KG1 leukemia cell line expressed very low levels of the protein, as analyzed by western blot (Fig. 7e, Supplementary Fig. 6) and flow cytometry (Fig. 7g). In vitro high-throughput drug sensitivity testing showed that U937 was much more sensitive to mitoxantrone and etoposide than KG1 (Supplementary Data 1). After selection of stably transfected KG1 cells overexpressing *SMARCA4*, confirmed by western blot of whole-cell lysates

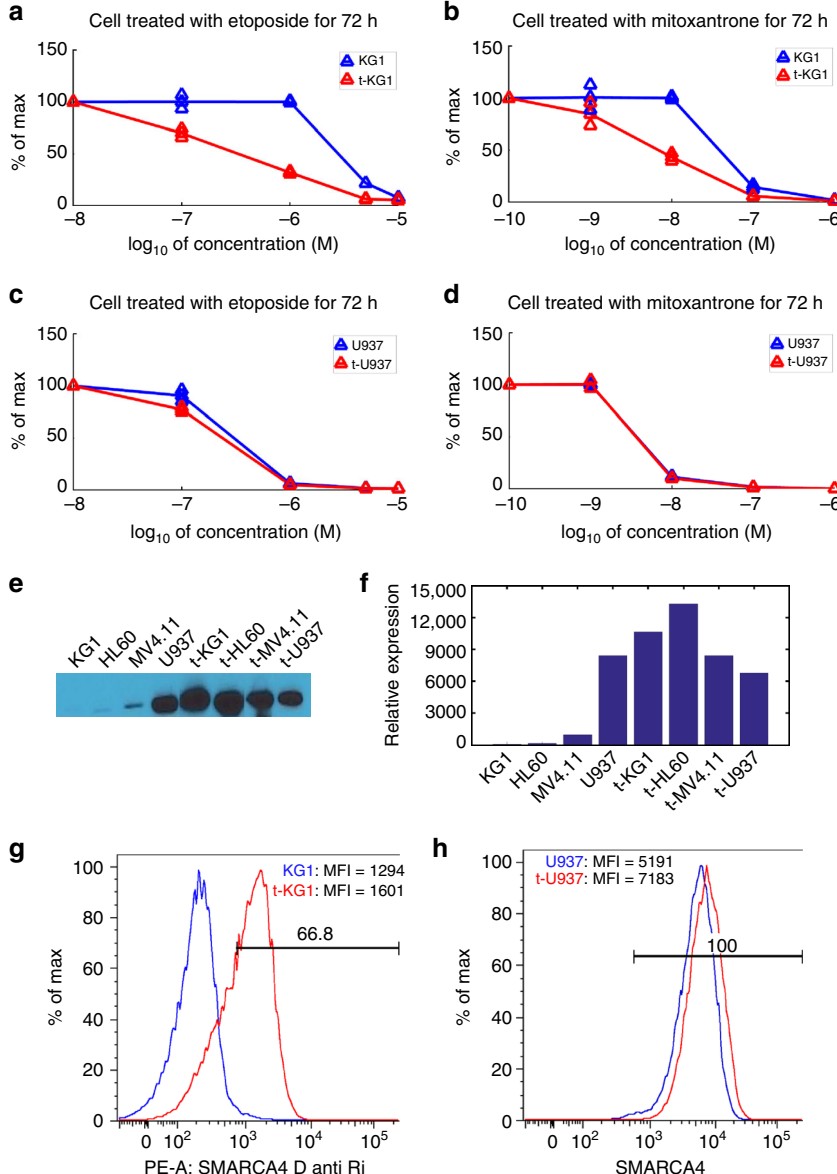

**Fig. 7** *SMARCA4* plasmid transfection experiments on cell lines KG1 and U937 for comparison of response to etoposide and mitoxantrone between original and transfected cells. **a**, **b** Comparison of the 72-h dose-response curves between KG1 cells (blue) and transfected KG1 cells (red) when cells are treated with (**a**) etoposide, and (**b**) mitoxantrone. **c**, **d** Comparison of the dose-response curves between U937 cells (blue) and transfected U937 cells (red) when cells are treated with (**c**) etoposide and (**d**) mitoxantrone. Three triangular marks at each point on the line indicate individual data points in duplicates and the average among them. The line connects averages of duplicates in each concentration measured. **e** Representative cropped western blot of control and transfected AML cell lines: KG1, U937, HL60, and MV4.11. Uncropped version is shown in Supplementary Fig. 6. **f** Quantifications of the SMARCA4 protein expression pattern of each of AML cell lines in (**e**). **g** Flow cytometry of *SMARCA4* surface expression data confirm the overexpression for KG1 (blue) vs. transfected KG1 (red). **h** Flow cytometry of *SMARCA4* surface expression data for U937 (blue) vs. transfected U937 (red). We note that U937 already strongly expresses *SMARCA4*, while KG1 exhibits minimal expression until after transfection. Abbreviations in **d** and **f** are as follows: PE-A, P-phycoerythrin area; MFI, mean fluorescence intensity; D anti Ri, Donkey anti-Rabbit

(Fig. 7e, Supplementary Fig. 6) and flow cytometry (Fig. 7g), in vitro chemo sensitivity testing demonstrated enhanced cytotoxicity (reduced viability) compared to non-transfected cells, more pronounced at the 72 h times (Fig. 7a, b) than 48 or 24 h (Supplementary Fig. 7) after addition of mitoxantrone or etoposide. Transfected U937 cells did not show significantly more sensitivity to these drugs compared to non-transfected cells (Fig. 7c, d), likely because U937 already showed a high expression of *SMARCA4*. Supplementary Table 4 summarizes the AUC and $IC_{50}$ values of transfected/non-transfected cells for each cell line and each drug. This result supports the hypothesis that *SMARCA4* expression drives sensitivity to etoposide and mitoxantrone.

We repeated these experiments using two more cell lines–HL60 and MV4.11–that show higher *SMARCA4* expression levels than KG1 but lower levels than U937 (Fig. 7f). HL60 shows lower *SMARCA4* expression than MV4.11; thus, we would expect to see more increased sensitivity in HL60 than in MV4.11 when overexpressing *SMARCA4*. As expected, we saw increased sensitivity to etoposide when HL60 was transfected to overexpress *SMARCA4* (Supplementary Fig. 8i) and no sensitivity change for MV4.11 (Supplementary Fig. 8k, l). HL60 had already been very sensitive to mitoxantrone before transfection to overexpress *SMARCA4*; thus, it was not surprising to see no change in sensitivity after transfection (Supplementary Fig. 8j). HL60 had

been as sensitive to mitoxantrone as U937, which is why overexpressing *SMARCA4* did not lead to a change in mitoxantrone response of HL60.

## Discussion

Due to the small sample size and the potential confounding factors in the gene expression and the drug sensitivity data, standard methods to discover gene-drug associations usually fail to identify replicable signals. We present a new way to identify robust gene-drug associations by prioritizing genes based on the multi-dimensional information on each gene's potential to drive cancer. We demonstrate that our method increases the chance that the identified gene-drug associations are replicated in validation data. This leads us to a short list of genes which are all attractive biomarkers for different classes of drugs. Our results— including the expression, drug sensitivity data, and association statistics from patient samples—have been made freely available to academic communities.

Our results suggest that high *SMARCA4* expression could be a molecular marker for sensitivity to topoisomerase II inhibitors in AML cells. These results offer a potentially enormous impact to improve patient response. Mitoxantrone is an anthracycline, like daunorubicin or idarubicin, and one of the two component classes of drugs included in nearly all upfront AML treatment regimens. It is also included (the "M") in the CLAG-M regimen[55], a triple-drug component upfront regimen now being studied as GCLAM[56]. Mitoxantrone and etoposide (also a topoisomerase II inhibitor) are two of the three drugs in the MEC regimen[57], used together with cytarabine, as a common regimen for relapsed/ refractory AML. Many modern regimens are in clinical trials that add an investigational drug to the MEC backbone, for example, an antibody to CXCR4 (NCT01120457) or an E selectin inhibitor (NCT02306291) in combination, or decitabine priming preceding the MEC regimen[58]. Identifying a predictor of response to mitoxantrone based on clinically available biospecimens, such as leukemic blast gene expression measured prior to treatment, could potentially increase median survival rates for patients with high expression of *SMARCA4* and indicate alternative therapies for patients with low *SMARCA4* expression.

The AML patients used in our study were consecutively enrolled on a protocol to obtain laboratory samples for research. They were selected solely based on sufficient leukemia cell numbers. As the patient samples were consecutively obtained and not selected for any specific attribute, we postulated that they were representative of patients seen at a tertiary referral center and that the results would be relevant to a larger, more general clinical population. Moreover, since each of the data sets from which we collected prior information (driver features) contained many more than 30 samples (e.g., TCGA AML data), it would be highly likely that MERGE results would be more generalizable to larger clinical populations than the methods that retrieve results specifically based on the 30 AML samples. In fact, Fig. 2a, b implies higher generalizability of MERGE compared to alternative methods.

While we have genotype information on *FLT3* and *NPM1* and the cytogenetic risk category for most of the 30 patients, the current version of the MERGE framework did not take these features into account: our main focus sought to build a general framework that could address the high-dimensionality challenge (i.e., the number of samples being much smaller than the number of genes) and make efficient use of expression data to identify robust associations. However, to consolidate our findings, we performed a covariate analysis to confirm that the top-ranked gene-drug associations discovered by MERGE remained significant when the risk group/cytogenetic features were considered

in the association analysis. We checked whether the gene-drug associations shown in the heat map in Fig. 6b (highlighted as red or green) were conserved when we added each of the following as an additional covariate to the linear model: (1) cytogenetic risk, (2) *FLT3* mutation status, and (3) *NPM1* mutation status. In Supplementary Fig. 9, each dot corresponds to a gene-drug pair, and each color to a different covariate. Most of the dots being closer to the diagonal indicates that the associations did not decrease significantly after adding the covariates. Moreover, of 357 dots, only eight were below the horizontal red line; this indicates that 98% of the gene-drug associations MERGE uncovered were still significant ($p \leq 0.05$) after modeling the covariate.

## Methods

**High-throughput drug sensitivity assay**. We developed a custom high-throughput screen for 160 oncology drugs, including a proportion that were FDA-approved (initially 40, now 62), with the remainder being investigational, that is, undergoing evaluation in cancer clinical trials. Only a limited number of drugs are FDA-approved for AML, and a majority are included in our assay, such as daunorubicin, idarubicin, cytarabine, mitoxantrone, thioguanine, arsenic trioxide, and midostaurin. Each drug was evaluated at eight different concentrations, in duplicate, and adherent to coated 384 well plates to mimic the marrow microenvironment, wherein attachment confers drug resistance. The drugs encompassed a broad range of classes with different mechanisms of action, including kinase inhibitors, nucleoside analogs, alkylating agents, mTOR inhibitors, anthracyclines, hypomethylating agents, steroids, and others (Supplementary Data 2). We profiled the drug sensitivity of an initial sample set of 30 viably cryopreserved primary patient AML specimens (24 new diagnosis, 6 relapsed/refractory) and 14 leukemia cell lines. A second validation set of 12 additional leukemia cell samples were studied from patients enrolled on a clinical trial for refractory AML (NCT02551718).

The patient samples, obtained with informed consent of patients on a protocol approved by the University of Washington-Fred Hutchinson Cancer Research Center Cancer Consortium IRB, were de-identified prior to study in the laboratory, with preservation of detailed clinical information, including: age, gender, cytogenetics, mutation status, antecedent hematologic disorder, initial blast count, initial platelet count, treatment regimen, response, and survival. Viably cryopreserved primary AML samples were thawed in the presence of DNase, then incubated for 48 h in IMDM containing 15% horse serum, 15% fetal calf serum, and very low level human stem cell factor (hSCF) (10 ng/ml). The cells were then subjected to density depletion on lymphocyte separation media and magnetic bead separation (Miltenyi MACS) if needed to prepare blast-enriched fractions with high (80–90%) viability and blast fractions exceeding 80%. Blast-enriched cell fractions were plated in 384 well plates coated with extracellular matrix protein at a density of 5000 cells/well in IMDM with 10% fetal calf serum. After plating overnight, drugs were robotically added at eight different concentrations spanning 4–5 log concentrations within a range of $10^{-12}$–$10^{-4}$ M, individually tailored for each drug. The samples were assayed in duplicate at each drug concentration. Viability was assessed after 4 days in culture using CellTiter-Glo Luminescent Cell Viability Assay (Promega, Madison, WI). These screens were performed at the Quellos HTS Core (http://depts.washington.edu/uwhts/) at the UW's Institute for Stem Cell and Regenerative Medicine (ISCRM). We compared duplicates of the AUC measure from curve fitting and observed a significant consistency (Pearson's correlation $r$: 0.94; $p$-value: 0) (Supplementary Fig. 10).

**The MERGE algorithm**. The MERGE algorithm takes the following data set: an expression data matrix $X$ ($p$ genes × $n$ patients), the drug response data matrix $Y$ ($q$ drugs × $n$ patients), and the driver feature matrix $D$ ($p$ genes × 5 driver features) (Fig. 1b, Supplementary Note 8). MERGE then learns the marker potential (MERGE score) of each gene, which is modeled as a weighted combination of the 5 driver features whose weights are data derived: MERGE learns these driver feature weights such that the resulting MERGE scores could explain observed gene-drug associations to the maximum extent possible. Genes with high MERGE score tend to be associated with more drugs than genes with low MERGE score (Fig. 1b).

To learn MERGE scores, we developed a probabilistic graphical model approach that provides a statistical model to represent relationships among variables in input data and a principled way of learning model parameters from data[59]. Probabilistic, model-based approaches have been proven to be a powerful tool to generate biological hypotheses from high-throughput molecular data[60–63]. The details on the probabilistic model used to develop the MERGE framework and its implementation are included in Supplementary Notes 13–15, Supplementary Figs. 11 and 12.

**Experimental validation for SMARCA4 in cell lines**. The leukemia cell lines were chosen for *SMARCA4* overexpression based on their relatively high and low sensitivity to mitoxantrone. The cell lines included KG1, U937, HL60, and MV4.11,

newly purchased for these experiments from the American Type Culture Collection (ATCC) (Manassas, VA) under a material transfer agreement. The cell lines were not re-authenticated, as they had just been received from the ATCC, nor were they tested for Mycoplasma given the short time from the time they were received from ATCC to use for these experiments. The cells were used at passage 3–9 for these experiments. The plasmid containing the cDNA for SMARCA4 and the Turbo-Fectin 8.0 reagent were obtained from Origene Technologies, Rockville, MD. The cell lines were transfected using the manufacturer's instructions. Successfully transfected cells were isolated after culture in 200 µg/ml G418. Clonal growing cells were obtained after 10 days, and these stable transfectants were used for subsequent chemo sensitivity assays. SMARCA4 protein expression was analyzed using a mouse monoclonal antibody (clone 6D7-F7-B6) against SMARCA4 (Origene Technologies, Rockville, MD) to stain western blots prepared by transfer from 4 to 20% sodium dodecyl sulfate polyacrylamide gels run on whole-cell lysates. The blots were incubated with the Pierce ECL western blotting substrate (Thermo Scientific, Rockford, IL). Band intensities were quantified using NIH ImageJ software. For chemotherapy sensitivity testing, cells were plated at 10,000 cells/well in a 96-well plate. The samples were assayed in duplicate at each drug concentration for cytotoxicity with mitoxantrone (0.001–1 µM) or etoposide (0.1–10 µM). Stably transfected and non-transfected cells were assayed in parallel. Analysis of cytotoxicity was performed by luminescent cell viability assay using CellTiter Glo (Promega, Madison, WI) after 24, 48, or 72 h after addition of chemotherapy.

**Code availability**. The implementation of the MERGE algorithm in MATLAB and R can be found in Methods section of the website associated with our study: http://merge.cs.washington.edu.

**Data availability**. The data generated and used in this study are provided as supplement. The genome-wide expression and in vitro drug sensitivity data from 30 AML patient samples and 14 AML cell lines are provided in Supplementary Data 1. The 160 drugs in our customized drug panel and their action mechanism classes are provided in Supplementary Data 2. The clinical information of the 30 AML patients is provided in Supplementary Data 3. The driver features collected to be used as input of the MERGE algorithm are provided in Supplementary Data 4. The processed RNA-seq and in vitro drug sensitivity data from 12 refractory AML patient samples used for validation are provided in Supplementary Data 5. All other relevant data are available upon request.

The genome-wide expression from our 30 patient samples used for discovery and 12 patient samples used for validation have also been deposited in Gene Expression Omnibus (GEO) under accession numbers GSE107465 and GSE108003, respectively. SuperSeries GSE108004 groups together GSE107465 (microarray) and GSE108003 (RNA-seq).

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

## Acknowledgements

This work was supported by the National Institutes of Health [T32 HL 007312]; National Science Foundation [DBI-1355899, and CAREER DBI-1552309]; American Cancer Society [127332-RSG-15-097-01-TBG]; Life Sciences Discovery Fund [2070888-01]; National Cancer Institute of the National Institutes of Health [P01CA077852]; and philanthropic funding from Norman Metcalfe. The content is solely the responsibility of the authors, and does not necessarily represent the official views of the funding agencies. We are grateful to Mohammad Javad Hosseini for useful discussions and performing some preliminary statistical experiments to evaluate our algorithm. We thankfully acknowledge receipt of one sample from the Fred Hutchinson Cancer Research Center/ University of Washington (FH/UW) Leukemia Repository.

## Author contributions

S.-I.L., S.Ce., V.G.O., C.A.B., and P.S.B. designed the study. S.-I.L. and S.Ce. developed the MERGE algorithm. S.Ce. performed all computational experiments except the DCS analysis performed by S.M.L. B.A.L. devised our way to narrow down the MERGE-prioritized associations and came up with the hypothesis on SMARCA4. T.J.M., C.A.B., V.G.O., C.P.M., and P.S.B. developed the high-throughput drug sensitivity assay. S.Ch. performed experiments for gene expression and drug sensitivity data generation. S.Ch. and J.D. carried out the overexpression studies for validation of the hypothesis on SMARCA4. E.H.E. provided the AML patient database of clinical features. A.S. gathered prior information on drugs and gene-drug associations. S.-I.L., S.Ce., and P.S.B. wrote the paper. P.S.B. wrote the sections pertaining to the AML cell preparation and drug sensitivity studies, and clinical relevance of the drug correlations. S.-I.L. and S.Ce. wrote the rest of the paper. All authors critically reviewed the paper.

## Additional information

**Competing interests:** The authors declare no competing financial interests.

