## [Peer Review File · Nature Communications]

Reviewers' comments:

Reviewer #1 (Remarks to the Author):

Review of: "An integrative framework for prioritizing candidate molecular markers reveals a novel driver for sensitivity to topoisomerase II inhibitors" submitted by Lee, et al.

In the submitted draft manuscript and submitted materials, the authors provide the following:

1. A new AML data resource in the form of:
 - a. Genome-wide expression profiles for 30 AML patients
 - b. Individual dose-response curves (obtained via in vitro studies) for the 30 patients and for a panel of 160 chemotherapy drugs and targeted agents.
 - c. Dose response curves for 14 AML cell lines for which expression data are publicly available (i.e., CGP and CCLE).

2. A novel heuristic that leverages prior information with the goal of providing a more powerful means for the identification of associations between gene expression and dose response.
 - a. The authors leverage prior information of the following types:
 - i. Mutation
 - ii. Copy number variation
 - iii. Methylation profiles (TCGA data)
 - iv. Expression profiles (from multiple studies)
 - v. Gene annotation databases
 - b. The authors provide the following evidence to support the utility of their heuristic:
 - i. A detailed comparison to prioritization of expression to
 1. simple linear correlation
 2. Elastic net [although no setting information appears to be provide]
 - ii. Split Sample Validation results using two disjoint subsets (each of size 12) of the 30 AML samples.
 - iii. A confirmatory investigation of data derived 14 AML cell lines.
 - iv. An examination of the consistency of identified gene by drug associations within drug functional classes.
 - v. Detailed biological interpretation of top ranked genes, and most notably, experimental validation of SMARCA4.

Summary of review:

In MERGE, the authors provide an interesting solution to a problem facing many researchers: how to leverage prior/external information to increase the power of possibly modest sized study. Central to the authors' approach is the penalized fitting of a simple linear model correlating drug response (as quantified by AUC) to gene expression. Their approach is to minimize an objective function (Suppl. Materials, page 5 Line 3) that has three terms: (1) a squared error term for the drug response as a linear function of gene expression, (2) a penalty term intended to shrink the linear coefficients of interest but in a preferential manner, and (3) a penalty term for driver feature weights. The second penalty term is the novel aspect of their proposed method. Rather than apply the same penalty to

all drug vs gene coefficients, the penalties vary with according to gene specific weights that are determined using information as described above in 2(a.). With respect to their final fitted model, the penalty corresponding to the inclusion of drug versus gene coefficients is intended to be inversely proportional to the strength of prior evidence regarding the biological/clinical relevance of the gene .

The proposed method, labeled MERGE, is an intuitively appealing approach by which information from other sources can be leveraged to improve inference on a study involving a modest size of samples. As such, and in general terms, their approach constitutes a nice addition to the existing literature. However, before the value of the submitted work can be accurately assessed, some clarifications are required. The requested clarifications are incorporated into the review comments provided below.

Specific Review Comments/Questions:

[1.] How many times were the validation analyses (e.g., the analysis involving two groups each of size 12 or the analysis of the 14 AML cell lines) performed in practice? Were there previous developmental versions of MERGE, and were they applied to the validation data until the current implementation of MERGE was arrived at? Or, are the validation studies "pristine" in the sense that the analyses were only performed once (i.e., using the final version of MERGE)? The manuscript would be improved with the inclusion, in the Supplementary Materials, of a discussion regarding how MERGE was developed and what steps were taken to make sure that none of the validation results were biased.

[2.] As per the description of the method provided above, I view your method as a LASSO inspired heuristic with a clever penalty mechanism that incorporates "prior knowledge". However, it appears to be presented in the manuscript as a Bayesian/Likelihood/Machine Learning heuristic. The following questions/comments pertain to Section 6 of the Supplementary Experimental Procedures (SEP). What is the general point of presenting the MERGE heuristic in terms of a parametric model? Is it to demonstrate that the heuristic can be thought of as being the natural extension of a reasonable Bayesian model parameterization of the problem under consideration? In its current form, the section does not provide sufficient details to support the claim on line 2 of page 5: "and this leads to the following optimization problem". Obviously, it appears that the result follows from assumptions concerning the normality of each of the components of the partitioned likelihood. However, the section should be modified to include clear specification of all the distributions involved. On line 17 of page 4, line 17 it is claimed that MERGE models the prior probability distribution over each w_{ij} . What of the V coefficients They are designated as having a probability distribution as well, yes? It appears then that W

and V are modeled in a Bayesian sense, i.e., that they are parameters for which belief in their values is modeled. Rather than employ traditional Bayesian methods, in which the posterior probability distributions are obtained, point estimates are arrived at via the minimization of the stated objective function. If this is so, it should be more clearly stated. As for the use of the term "learns" (line 25 of page 4), I would suggest re-stating as "MERGE selects driver feature weights such that...".

[3.] A section should be included in the Supplementary Experimental Procedures that details the computational aspects of minimizing the objective function provided in Eq (2) on page 5. To what extent does the parameterization provide an identifiable model? How confident are the authors that the global minimum was identified? What type of computing power was required to perform their MERGE analysis? What were the exact cross validation settings? The fact that $C=100$ was selected and is the smallest of the considered values begs the question as to why smaller values of C weren't considered? What values for λ and C were utilized in the split sample validation study (i.e., involving two groups of size 12 each)? Which is to say, were λ and C values obtained via CV for both of those sets and, if so, to what extent did the reduced sample size (i.e., 12 compared to 30) present any complications?

[4.] What was the dissimilarity metric and clustering method used for the agglomerative clustering presented in Figure 3A? They should be stated explicitly. Could the authors clarify how the Fisher's Exact Test was performed with respect to Figure 3A? Can the authors provide (in the response to this review) the actual tabled counts used?

[5] It appears that Figure 3B provides a plot of the observed DCS quantiles versus the averaged quantiles across 1000 permutations. Was the reported p-value adjusted against the p-value distribution corresponding to the difference (as quantified by the KS p-value) between the quantiles of each permutation and the average quantiles?

[6] A selection method based upon Pearson's Linear Correlation is provided. Was a method based upon Spearman's Rank Correlation considered? Spearman's seems preferable to Pearson's as it detects monotonic, not just linear, relationships and is more robust to outliers.

[7] The poor performance of the ElasticNet is surprising. If I understand the results correctly, for smaller values of N , ElasticNet performs horribly; especially compared to random selection (as represented by grey lines). The poor performance of in Figure 2A is very troubling. In the supplementary materials, the authors should provide exact details with respect to the implementation of ElasticNet on this data. The results in Figure 2A require some type of explanation, a closer examination of the results and at least a plausible hypothesis as to the ElasticNet's failure. Could it be possible that the method by which

the top

53N top genes are selected after the ElasticNet fit is providing misleading results? Did the ElasticNet ever provide a number of non-zero selected genes such that the number was less than the desired number of top genes to be selected?

[8] A minor point: Figure 5A appears to visualize the \log_{10} p-values rather than the $-\log_{10}$ p-values.

[9] A minor point: Figure 4A could be inset into the white space of Figure 4B to conserve space.

[10] At the minimum, the Supplementary Materials should include Figures such as 5A and 5B but corresponding to genes selected under Pearson's Linear Correlation (and Spearman's as well, possibly) and the ElasticNet. Does MERGE truly provide a path towards identifying the roles (as detailed on pages 8 and 9) of FLT3, CASP8AP2, L2HGDH, MNT, BAZ2B, MZF1, BEX2, and SMARC4 that other methods would not have?

[11] On line 6 of page 3 of Suppl. Materials, the authors state that the driver features were standardized. How were the "Candidate regulator" and CNV scores standardized? Are those driver features purely binary?

[12] The discussion section would be improved by including some comments concerning the demographic and clinical attributes of the 30 AML patients from which the sample materials were obtained. What larger, more general clinical population, can the results be postulated as being relevant to? Does the inclusion of "prior" information from cell lines and other sources, suggest that it is possible that the results could be considered more generalizable than those obtained by an analysis only incorporating the data from the 30 AML samples?

[13] An experimental design involving a truly independent validation arm (e.g., involving an additional 30 AML patients) would have dramatically improved the potential impact of the work.

[14] By what method was the FDR estimated in the various components of the analysis?

[15] Did authors consider summary methylation and CNV measures other than sample mean? For methylation profiles, in particular, was there any concern that averaging across all CpG results (mapped to a given gene) might be less efficient than a feature selected (and hopefully more informative than the average) subset of CpGs?

Reviewer #2 (Remarks to the Author):

This is an interesting and technically strong paper to identify gene expression markers of drug response in acute myeloid leukemia for many classes of therapeutics. The study contributes both a valuable data set — namely gene expression data for 30 AML patient-derived specimens, together

with dose response curves in these samples measured for 160 drugs — and a simple but principled approach to incorporate prior knowledge about the gene relevance into the computational model for identifying gene-drug associations, which is called the MERGE algorithm. The MERGE optimization problem modifies a standard regression approach for predicting drug responses from gene expression values by simultaneously learning a common weighting over 5 gene features (mutations in TCGA/AML, expression “hubness”, “regulator” status, CNVs from TCGA/AML, methylation from TCGA/AML); the weighted features or MERGE score appears in a weighted L2 regularization function in the joint learning of regression models for all the drugs, enforcing a shared set of prior on relevant drugs. The authors show that MERGE improves consistency of gene-drug associations between training and test sets compared to standard elastic net and identifies known or plausible associations for several classes of drugs. They also provide experimental validation for a novel association between expression of SMARC4 and response to topoisomerase II inhibitors by showing that overexpression of SMARC4 in a leukemia cell lines with low/medium endogenous expression tends to sensitize cells to these drugs (though some of these cell lines are already sensitive to specific topoisomerase II inhibitors without SMARC4 overexpression).

Overall this is a solid paper that provides some evidence that the method can find novel gene-drug associations that may be worth exploring further. The main weakness is that further method comparison and statistical assessment is needed, since there has been much recent work on predicting drug response from molecular data in cell lines. Specific additional analyses are suggested below:

(1) The method is evaluated based on consistency of gene-drug associations between training and test sets and compared to elastic net and a simple correlation statistic by this measure. However, the MERGE algorithm actually solves a regression problem, and most work in this area trains predictive models and assesses regression performance on the test set. Here the data sets are all small (30 new patient-derived samples, 12 cell lines in CCLE), but it would still be helpful to try to assess whether MERGE outperforms elastic net as a regression model on held out cell lines, or is the advantage only for expression marker identification?

(2) There are a number of other recently published methods for learning to predict drug sensitivity, and it would be informative to compare these methods to MERGE in terms of both prediction accuracy and consistency of gene-drug associations. Of particular interest are multi-task methods, since the MERGE score-based L2 regularizer shares information across the drug tasks. It would be straightforward to compare to the recently published trace norm multi-task drug prediction approach (Yuan, Paskov et al, Nature Sci Reports 2016). Another option is the Bayesian multitask kernel method that was the winner in a recent DREAM competition (Costello et al., Nat Biotechnol 2014), though as a kernel method, this approach gives prediction performance more readily than gene-drug associations.

(3) FLT3 mutation status in individual cell lines is not used in the MERGE model, although patient mutation status was collected. For many reasons both practical and historical, patient mutation status is far more likely than expression level to be measured and used for making therapeutic decisions. The authors state that their finding of an association between FLT3 gene expression level and drug response for multiple agents (including FLT3 inhibitors) is novel. It would be valuable to explore this issue more deeply — at least to quantify whether the gene expression level gives a more statistically significant marker of response than mutation status, at least in the drug response assays.

Minor issues:

- Figure 7 seems like it could be concatenated with Figure 6 or moved to the supplement — the panels are just confirming expression/overexpression of SMARC4 and not describing drug response validations

- "the repertoire of drugs for cancer is rapidly expanding" — this statement in the introduction is not generally considered to be true; rather, it is expensive to develop a new drug and easier to repurpose an existing agent
- "which is a more challenging setting and the testability" — non-grammatical statement, and in general the "testability" language is confusing
- DNase rather than DNase

Reviewer #3 (Remarks to the Author):

Manuscript by Lee et al presents data from gene expression and drug screening of 30 AML patients and 14 AML cell lines. Authors propose use of MERGE a computational method that uses genes biological importance/function and identified gene expression signatures for drug response. Authors propose that using multi-dimensional information on a gene helps in determine its potential as a cancer driver which can impact its influence on drug sensitivity. Overall the study is novel and nicely designed however there are some major concerns as listed below:

1. Results: Author indicate that 12 of 15 drugs that are used in clinic for AML treatment were associated with CR, one of the most commonly used drug cytarabine is missing from this list, authors should provide some comments on this. Authors should provide information on of the drugs with significant association between drug sensitivity and complete remission how many agents were actually used to treat the cohort of 30 patients
2. Although authors point out that MERGE algorithm uses relative importance of each driver feature details on this are lacking and should be provided. Similarly criteria for driver feature characterization should be provided.
3. Authors should also provide information of risk group category/ cytogenetic features for these 30 patients and whether or not these were taken into account in the association analysis

Reviewer 1:

Review of: “An integrative framework for prioritizing candidate molecular markers reveals a novel driver for sensitivity to topoisomerase II inhibitors” submitted by Lee, et al.

In the submitted draft manuscript and submitted materials, the authors provide the following:

- 1. A new AML data resource in the form of:*
 - a. Genome-wide expression profiles for 30 AML patients*
 - b. Individual dose-response curves (obtained via in vitro studies) for the 30 patients and for a panel of 160 chemotherapy drugs and targeted agents.*
 - c. Dose response curves for 14 AML cell lines for which expression data are publicly available (i.e., CGP and CCLE).*

2. A novel heuristic that leverages prior information with the goal of providing a more powerful means for the identification of associations between gene expression and dose response.
 - a. The authors leverage prior information of the following types:
 - i. Mutation
 - ii. Copy number variation
 - iii. Methylation profiles (TCGA data)
 - iv. Expression profiles (from multiple studies)
 - v. Gene annotation databases
 - b. The authors provide the following evidence to support the utility of their heuristic:
 - i. A detailed comparison to prioritization of expression to
 1. simple linear correlation
 2. Elastic net [although no setting information appears to be provide]
 - ii. Split Sample Validation results using two disjoint subsets (each of size 12) of the 30 AML samples.
 - iii. A confirmatory investigation of data derived 14 AML cell lines.
 - iv. An examination of the consistency of identified gene by drug associations within drug functional classes.
 - v. Detailed biological interpretation of top ranked genes, and most notably, experimental validation of SMARCA4.

Summary of review:

In MERGE, the authors provide an interesting solution to a problem facing many researchers: how to leverage prior/external information to increase the power of possibly modest sized study. Central to the authors' approach is the penalized fitting of a simple linear model correlating drug response (as quantified by AUC) to gene expression. Their approach is to minimize an objective function (Suppl. Materials, page 5 Line 3) that has three terms: (1) a squared error term for the drug response as a linear function of gene expression, (2) a penalty term intended to shrink the linear coefficients of interest but in a preferential manner, and (3) a penalty term for driver feature weights. The second penalty term is the novel aspect of their proposed method. Rather than apply the same penalty to all drug vs gene coefficients, the penalties vary with according to gene specific weights that are determined using information as described above in 2(a.). With respect to their final fitted model, the penalty corresponding to the inclusion of drug versus gene coefficients is intended to be inversely proportional to the strength of prior evidence regarding the biological/clinical relevance of the gene.

The proposed method, labeled MERGE, is an intuitively appealing approach by which information from other sources can be leveraged to improve inference on a study involving a modest size of samples. As such, and in general terms, their approach constitutes a nice addition to the existing literature. However, before the value of the submitted work can be accurately assessed, some clarifications are required. The requested clarifications are incorporated into the review comments provided below.

Specific Review Comments/Questions:

[1.] How many times were the validation analyses (e.g., the analysis involving two groups each of size 12 or the analysis of the 14 AML cell lines) performed in practice?

Were there previous developmental versions of MERGE, and were they applied to the validation data until the current implementation of MERGE was arrived at? Or, are the validation studies “pristine” in the sense that the analyses were only performed once (i.e., using the final version of MERGE)?

The manuscript would be improved with the inclusion, in the Supplementary Materials, of a discussion regarding how MERGE was developed and what steps were taken to make sure that none of the validation results were biased.

We thank the reviewer for raising a great point. It is a valid point that the model development and improvement process should be considered as part of the “training” phase. Therefore, when assessing the performance of the proposed approach by comparing it with existing approaches, we should use a separate, validation dataset that was not used for the model development and improvement steps.

The MERGE algorithm presented in the original manuscript was our first and last version of the algorithm for learning genes’ priority scores based on the driver potential of the genes. We did not modify the MERGE algorithm such that it shows a better validation performance on any of our data from 30 patient specimens or 14 AML cell lines. As we described in the summary #2 above, we made a minor modification by removing the regularization term with the hyperparameter C , which led to the final version of the MERGE algorithm presented in the revised manuscript. The results are almost identical to those in the original manuscript. Since we did not decide to remove this term based on the validation results, these validation studies are not biased. We want to reassure the reviewer that when we chose the values of the hyperparameter λ , we used the training portion of the data to learn these hyperparameters. As the reviewer suggested, we incorporated into the manuscript a discussion on how we developed MERGE before measuring its performance on the validation data (SI Page 6, Line 11).

Additionally, to further remove the concern raised by the reviewer, we generated 12 additional samples and used them as additional validation data for the consistency of significant gene-drug associations. We have not modified the MERGE algorithm after getting the validation results on this new dataset from 12 samples. We show in Figure 2C that MERGE outperforms 4 other approaches when tested on this new dataset. We incorporated these results into the revised manuscript (Page 5, Line 35).

[2.] As per the description of the method provided above, I view your method as a LASSO inspired heuristic with a clever penalty mechanism that incorporates “prior knowledge”. However, it appears to be presented in the manuscript as a Bayesian/Likelihood/Machine Learning heuristic. The following questions/comments pertain to Section 6 of the Supplementary Experimental Procedures (SEP).

What is the general point of presenting the MERGE heuristic in terms of a parametric model? Is it to demonstrate that the heuristic can be thought of as being the natural extension of a reasonable Bayesian model parameterization of the problem under consideration?

We thank the reviewer for the question. Perhaps, our description of the MERGE model was not clear in the original manuscript. We view MERGE not as a LASSO inspired heuristic, but as a *probabilistic model-based approach* which has been proven to be a powerful tool to generate biological hypotheses from high-throughput molecular data [1-3]. A probabilistic method has several important advantages. First, it provides an expressive model to describe relationships among variables. Second, these learned probabilistic relationships can be read off from the learned model, which often directly lead to comprehensive biological interpretation. In MERGE, the prior variance of the gene-drug weights is

interpreted as the corresponding gene's *biomarker potential*. We clarified that MERGE is a probabilistic model-based approach in the Supplemental Experimental Procedure 11 of the revised manuscript (SI Page 6, Line 16).

[1] Segal E, Shapira M, Regev A, Pe'er D, Botstein D, Koller D, Friedman N. Module networks: identifying regulatory modules and their condition-specific regulators from gene expression data. *Nature Genetics* 2003 Jun;34(2):166-76.

[2] Akavia, U.D.*, Litvin O.*, Kim J., Sanchez-Garcia F., Kotliar D., Causton H.C., Pochanard P., Mozes E, Garraway L.A., Pe'er D. An Integrated Approach to Uncover Drivers of Cancer. *Cell* 2010; 143:1005-1017.

[3] Celik S, Logsdon B, Battle S, Drescher C, Rendi M, Hawkins D., Lee S.-I. Extracting a low-dimensional description of multiple gene expression datasets reveals a potential driver for tumor-associated stroma in ovarian cancer. *Genome Medicine* 2016 Jun 10;8(1):66.

In its current form, the section does not provide sufficient details to support the claim on line 2 of page 5: "and this leads to the following optimization problem". Obviously, it appears that the result follows from assumptions concerning the normality of each of the components of the partitioned likelihood. However, the section should be modified to include clear specification of all the distributions involved.

We thank the reviewer for the suggestion. We improved Supplemental Experimental Procedure 10 ("MERGE algorithm" section) by describing the details about the joint probability distribution and its decomposition to conditional distributions, specifications of these conditional distributions, the log-likelihood function and the derivation of the optimization problem (SI Page 5, Line 13).

On line 17 of page 4, line 17 it is claimed that MERGE models the prior probability distribution over each w_{ij} . What of the V coefficients They are designated as having a probability distribution as well, yes? It appears then that W and V are modeled in a Bayesian sense, i.e., that they are parameters for which belief in their values is modeled. Rather than employ traditional Bayesian methods, in which the posterior probability distributions are obtained, point estimates are arrived at via the minimization of the stated objective function. If this is so, it should be more clearly stated.

We thank the reviewer for the insightful question and suggestion. As the reviewer pointed, we modeled W and V in a Bayesian sense (i.e., they are parameters for which belief in their values is modeled), where W is modeled as a Gaussian random variable whose variance is modeled based on V , and V is modeled as a uniform random variable (i.e., $P(V)$ is constant). As the reviewer pointed out, instead of a traditional Bayesian approach (i.e., estimating $P(\Theta | \mathbf{D})$, the full posterior distribution over Θ), we employed *maximum a posteriori probability (MAP) estimation* (i.e., obtaining a point estimate of Θ that maximizes $P(\Theta, \mathbf{D})$). MAP estimation has two advantages over the traditional Bayesian approach. First, estimating specific parameter values makes biological interpretation straightforward. For example, specific parameter values of v_k coefficients enable an efficient computation of the MERGE scores and straightforward interpretation of how driver features impact the MERGE score (i.e., biomarker potential). Second, MAP estimation allows a much simpler learning procedure of the parameters, especially when $P(\Theta | \mathbf{D})$ does not have a closed-form solution. Penalized linear regression models such as LASSO (or RIDGE) also employs MAP estimation for a probabilistic model (specifically, linear regression model) with a laplacian (or Gaussian) prior for $P(\Theta)$, where the parameter Θ means the W values. MERGE extends the penalized linear regression models by modeling the variance of the W parameters based on V

and the driver features of genes. *To summarize, MERGE is a probabilistic model-based approach that uses the MAP estimation to estimate the parameters.*

In response to the reviewer's comment, we significantly improved Supplemental Experimental Procedure 10 ("MERGE algorithm" section) and also added Supplemental Experimental Procedure 11 where we described the probabilistic model of MERGE in more detail and its learning algorithm to address the points above in the revised manuscript (SI Page 6, Line 24).

As for the use of the term "learns" (line 25 of page 4), I would suggest re-stating as "MERGE selects driver feature weights such that..."

We thank the reviewer for the suggestion. However, we believe that the term "estimate" or "learn" is more commonly used than "select" when we refer to the process of obtaining the parameter values that minimize a probabilistic objective function, as we do in Eq (2). The term "select" is more commonly used when we choose a subset of features, for example, by using a sparsity prior such as in LASSO. We hope that our revision clarifies that MERGE is a probabilistic model-based approach with an objective function to optimize, and that the use of the term "learns" is more natural in the revised manuscript. Thanks to the reviewer's comment, we now have a much improved description of the probabilistic model and the optimization problem.

[3.] A section should be included in the Supplementary Experimental Procedures that details the computational aspects of minimizing the objective function provided in Eq (2) on page 5.

We thank the reviewer for the comment, which allowed us to improve our manuscript by clarifying the computational aspects of minimizing the objective function of MERGE. In response to the reviewer's comments below, we improved Supplemental Experimental Procedure 10 ("MERGE algorithm" section) and also added Supplemental Experimental Procedures 11, 12 and 13 which detail the computational aspects of MERGE. We responded in detail to the comments of the reviewer below.

To what extent does the parameterization provide an identifiable model?

The objective function is a non-convex function; thus different initializations of the V vector (v_k values for each driver feature k) may lead to different learned parameters, i.e. different local minima of the objective function in Eq (2) in the Supplemental Experimental Procedure 10. Practically, however, we observed that when we tried different initializations of V (and correspondingly W), the learned parameters were very similar to each other (more details below). In our application of MERGE in the current manuscript, we initialized all five v_k values to be zero because that gives us an unbiased starting point -- an equal prior variance for the weight values of all genes. Below, we describe our results on the consistency between the zero initialization and random initializations. In the formal sense of model identifiability the learning parameters (v_k 's and w_{ij} 's) of interest become perfectly identified in the limit of an infinite number of individuals. We added Supplemental Experimental Procedure 12 to the revised manuscript, where we discuss MERGE model's identifiability (SI Page 6, Line 40).

How confident are the authors that the global minimum was identified?

We thank the reviewer for pointing this out. We believe that the global minimum was identified because many random initializations converged to roughly the same point.

The objective function of MERGE is not jointly convex wrt V and W , though it is convex wrt each set of parameters with the other set held fixed. Given V held fixed, the objective is convex wrt W , and given W held fixed, the objective is convex wrt V . This means that each learning step in the block coordinate descent algorithm, learning V or learning W , is a convex optimization problem (SI Page 6, Line 4).

In theory, for non-convex objective, different initializations would result in different learned parameters (i.e., different local minima). In practice, however, depending on the objective function and the input data, it is possible that a roughly unique solution is identified, empirically. One way to check is to try multiple runs with different initializations of parameters and see whether these runs converge to roughly the same point.

To address the reviewer's comment, we performed 20 different runs of MERGE where we initialized v_k values such that they are generated randomly from a standard normal distribution. Then we compared the resulting training objective function values and the MERGE scores from these 20 runs to those from the MERGE run used in our paper (i.e., where we initialized all five v_k values to be zero). We performed this experiment with the same hyperparameter value selected by LOOCV and used for the final model ($\lambda = 20$) (Figure S7A) and with one more λ value ($\lambda = 50$) (Figure S7B). As can be seen in the top portion of Figure S7A, for $\lambda = 20$, only 4 out of 20 runs resulted in a smaller objective function (i.e. better local optima) compared to our initialization with zero v_k values, although the difference was very small.

All the 20 runs with random initializations resulted in very similar objective function values (Figure S7A-top), almost the same v_k values shown in Figure 4, and exactly the same gene rankings as those from the zero initialization of MERGE (Figure S7A-bottom). This indicates that different random initializations converged to roughly the same point. A different value of $\lambda = 50$ showed consistent patterns (Figure S7B). These results are described in the revised manuscript (SI Page 7, Line 6).

What type of computing power was required to perform their MERGE analysis?

MERGE does not require any special computing power. We ran MERGE using R (version 3.3.2) on a machine with an Intel(R) Xeon(R) E5645 2.40GHz CPU and 24GB RAM. One MERGE run on the data with ~17K genes, 53 drugs and 30 samples took 12 seconds on that machine. We indicated that in the revised manuscript (SI Page 6, Line 8).

What were the exact cross validation settings?

Cross validation experiments were performed when measuring the prediction accuracy on left-out, test data (Figure 7) and when selecting the values of the hyperparameter (regularization tuning parameter λ). To select the λ value, we performed the leave-one-out cross validation (LOOCV) test within the training data, in order to choose one λ value out of the 19 λ values in a wide range [1,100]. We used LOOCV tests to choose the values of the tuning parameters for other methods as well: elastic net regression, multitask learning method and the DREAM challenge winner Bayesian multi-task multiple kernel learning method. We added a section that clarifies the details on cross validation tests to the revised manuscript (SI Page 7, Line 19).

There are largely three settings where we used cross-validation tests.

- 1) Measuring the prediction accuracy via LOOCV (Figure 7B): In each fold, in which we left one sample out and used the rest of the ($n-1$) samples to train the model, we performed the "inner loop" LOOCV

using those ($n-1$) samples to choose the λ value and trained the model using ($n-1$) samples using the chosen λ value.

- 2) Training the model using one 12-sample batch and testing on the other batch (Figure 2A, Figure 7A): We performed LOOCV tests within 12 samples in each batch to choose the value of the tuning parameter and train the model. Then, we tested the consistency of significant gene-drug associations (Figure 2A) and prediction accuracy measured by rank-correlation of samples (Figure 7A).
- 3) Training the model using all 30 samples (Figure 2B-2C): We first performed the LOOCV tests to choose the λ value using all 30 samples and then trained the model using all samples with the chosen λ value.

The fact that $C=100$ was selected and is the smallest of the considered values begs the question as to why smaller values of C weren't considered?

We thank the reviewer for making an excellent point. As was shown in Figure S1 of the original manuscript, varying the value of C did not lead to a significant variation in the mean squared error, especially between $C=0$ and $C=100$. Then we realized that this may be because the 3rd term in the objective function, Eq (2) (SI Page 5, Line 28), has an effect to regularize the v_k values. This implies that we do not need the L2 regularization term on v_k values which involves the hyperparameter C . For this reason, we modified the MERGE objective function such that it now contains only one hyperparameter λ , which resulted in a simpler model and a more efficient learning algorithm. Not surprisingly, there is almost no impact on the results caused by the exclusion of C (given that the C value originally used was small.)

We extended Supplemental Experimental Procedure 10 (“MERGE algorithm” section) such that it now describes the probabilistic justification of the MERGE model formulation in greater detail. We reselected the hyperparameter λ via LOOCV to be 20 (similar to the previous model) and reran MERGE based on the improved model formulation.

What values for lambda and C were utilized in the split sample validation study (i.e., involving two groups of size 12 each)? Which is to say, were lambda and C values obtained via CV for both of those sets and, if so, to what extent did the reduced sample size (i.e., 12 compared to 30) present any complications?

Selecting the hyperparameter values is part of the model training process and therefore, we should use only training data for that. In the split sample validation study, this means that we should choose the λ value via LOOCV for each of the 12-sample batches separately, as described above in 2). Among the 19 λ values in the range [1, 100], the selected λ was 30 for batch 1, 9 for batch 2, and 20 for all 30 samples.

We agree with the reviewer that a different sample size of data can impact the selected λ value. Here is why. It is well-known that when the sample size is too small, regression methods are more prone to overfitting (i.e., failure in learning a model that is generalizable to unseen test data). In that case, a regularized model would tend to choose a large λ value via cross-validation within training data in order to prevent overfitting. However, the selected λ value can be affected by other factors as well, as exemplified below.

At the first glance, our observation in the split sample test was consistent with what is stated above. The selected λ for the 12 samples in batch 1 ($\lambda=30$) is larger than the selected λ for all 30 samples ($\lambda=20$), as

expected due to the small sample size. However, although both batch 1 and batch 2 have 12 samples, the selected λ for batch 2 ($\lambda=9$) is much smaller than the selected λ for batch 1 ($\lambda=30$). We conjecture that the reason is more overfitting happening in batch 2, as implied by the high intra-batch similarity in batch 2 compared to batch 1 (t -test p : $3.8e-2$ for drug response correlations, and p : $5.5e-2$ for expression level correlations). Consistently, we observed that both gene-drug association consistency and drug response prediction performance of the model trained in batch 2 were worse than the models trained in batch 1. When the sample size is small, the training data is also more likely to have a lower variation than when the sample size is large. This is likely to lead to more overfitting sometimes and even with the LOOCV, the regularized method may fail to select the optimal regularization parameter λ .

In any statistical model, a small sample size often leads to overfitting and high variability. To alleviate these problems, we used LOOCV (instead of, e.g., 5-fold CV) to select λ value and averaged results between 2 batches in the split sample validation study (training with batch 1 and testing on batch 2, and vice versa). We discussed the potential issues with the small sample size in the revised manuscript (SI Page 7, Line 43).

[4.] What was the dissimilarity metric and clustering method used for the agglomerative clustering presented in Figure 3A? They should be stated explicitly.

We thank the reviewer for the suggestion. To generate Figure 3A, we used the Euclidean distance as the dissimilarity metric and average linkage as the clustering method. We incorporated this information into the revised manuscript (SI Page 4, Line 5).

Could the authors clarify how the Fisher's Exact Test was performed with respect to Figure 3A?

We thank the reviewer for this question. In the dendrogram, for each internal node that has 3 to 8 drugs in their leaves, we first retrieved a unique set of drug classes that contain at least one drug from the node's downstream drugs. Then, we checked the enrichment of each of these drug classes in the set of the downstream drugs of that node. The presented p -values in the original manuscript were the nominal p -values from Fisher's exact test (they were not corrected for multiple hypothesis testing). In the revised manuscript, we updated Fig 3A such that each p -value we present is FDR-corrected, using the method introduced by [1] referenced in our response to R1's comment #14 below, for the number of drug classes tested for that node. We incorporated this detail into the revised manuscript (SI Page 4, Line 7).

Can the authors provide (in the response to this review) the actual tabled counts used?

We listed below the numbers of drugs in two sets (a drug class, and downstream drugs) and their intersection used to get the FDR-corrected Fisher's exact test p -values presented in Fig 3A. The total number of drugs is 53. The p -value shown below are FDR-corrected.

mTOR inh (p: $4.2e-6$): #drugs in the class=6, #drugs in the node=5, #drugs in intersect=5
AKT/PI3K inh (p: $2.5e-4$): #drugs in the class=6, #drugs in the node=5, #drugs in intersect=4
Kinase inh (p: $6e-2$): #drugs in the class=5, #drugs in the node=8, #drugs in intersect=3
Flt3 inh (p: $6e-2$): #drugs in the class=5, #drugs in the node=4, #drugs in intersect=2
Bcl2 inh (p: $1.3e-2$): #drugs in the class=3, #drugs in the node=3, #drugs in intersect=2
Nucleoside anlg (p: $6.3e-3$): #drugs in the class=8, #drugs in the node=6, #drugs in intersect=4
Topoisom inh (p: $8.9e-3$): #drugs in the class=8, #drugs in the node=4, #drugs in intersect=3

Microtbl inhb (p: 1.7e-3): #drugs in the class=5, #drugs in the node=3, #drugs in intersect=3
HDAC inhb (p: 3.4e-6): #drugs in the class=4, #drugs in the node=4, #drugs in intersect=4
GSK-3 inhb (p: 2.2e-2): #drugs in the class=4, #drugs in the node=2, #drugs in intersect=2

[5] It appears that Figure 3B provides a plot of the observed DCS quantiles versus the averaged quantiles across 1000 permutations. Was the reported p-value adjusted against the p-value distribution corresponding to the difference (as quantified by the KS p-value) between the quantiles of each permutation and the average quantiles?

We thank the reviewer for the excellent point. The reported KS test p-value was not adjusted using the empirical distribution of differences from the average over the 1,000 permutations. To address this and keep things simple, we created a QQ plot of the empirical DCS score p-values from 1,000 random permutation tests against the Unif(0,1) distribution quantiles (rather than the average quantiles) for each gene. We performed this for all genes that had significant association with at least 1 drug. If the DCS scores were random, then the empirical DCS score p-values should have followed a Unif(0,1) distribution, but the updated Figure 3B shows that we get more significant p-values than expected by chance. The new p-value of 0.029 is adjusted against the p-value distribution corresponding to the difference (as quantified by the KS test p-value) between the quantiles of each permutation and the Unif(0,1) distribution quantiles. We incorporated this result into the revised manuscript (Page 6, Line 37).

[6] A selection method based upon Pearson's Linear Correlation is provided. Was a method based upon Spearman's Rank Correlation considered? Spearman's seems preferable to Pearson's as it detects monotonic, not just linear, relationships and is more robust to outliers.

We thank the reviewer for the great suggestion. We added the results from the Spearman correlation to Figure 2 and Figure 3C.

[7] The poor performance of the ElasticNet is surprising. If I understand the results correctly, for smaller values of N, ElasticNet performs horribly; especially compared to random selection (as represented by grey lines). The poor performance of in Figure 2A is very troubling. In the supplementary materials, the authors should provide exact details with respect to the implementation of ElasticNet on this data.

We provided the details on the ElasticNet implementation in Supplemental Experimental Procedure 14. We are using ElasticNet from the *glmnet* R package on CRAN. In the original manuscript, we only standardized (i.e., centered and scaled to have mean 0 and standard deviation 1) the predictors, and did not standardize the responses for ElasticNet. To make all methods directly comparable to MERGE, we standardized both the predictors and the responses, which led to a slight improvement in ElasticNet's performance (specifically for $x=[0, 1000]$ in Figure 2A). However, it still underperforms when compared to the random curves in Figure 2A, as the reviewer pointed out.

We described in detail below the potential reasons for the poor performance of ElasticNet.

The results in Figure 2A require some type of explanation, a closer examination of the results and at least a plausible hypothesis as to the ElasticNet's failure.

We thank the reviewer for the comment that would help us explain the poor performance of ElasticNet and multi-task learning method. There are multiple reasons for the poor performance of ElasticNet in

Figure 2A. First, ElasticNet (and the newly added multi-task learning method) is a multiple regression method intended to solve a prediction problem, and it is not specifically designed for capturing robust gene-drug associations. On the other hand, MERGE was designed to aggressively decrease the number of false positive gene-drug associations by incorporating prior knowledge on the genes' potential to drive the disease. Secondly, since this problem is ultra high-dimensional (with data consisting of 12 samples and ~17K variables for Figure 2A), multiple regression methods are likely to learn too complex models to identify robust gene-drug associations, even with regularization (e.g., elastic net penalty). Finally, it is well-known that when there are highly correlated features, sparse regression methods -- such as LASSO and elastic net regression -- (arbitrarily) select one of these features and almost exclude the other correlated features. Many robust gene-drug pairs whose associations would have been replicated in validation data are likely to have been eliminated that way. Figure S9 shows that many of the highly correlated gene-drug pairs do not learn strong weights in the multiple regression models. We described potential reasons for the poor performance of ElasticNet in the revised manuscript (Page 6 Line 8).

Could it be possible that the method by which the top 53N top genes are selected after the ElasticNet fit is providing misleading results? Did the ElasticNet ever provide a number of non-zero selected genes such that the number was less than the desired number of top genes to be selected?

We thank the reviewer again for the great point. Here is how we identified the 53 N genes for ElasticNet. We first ordered the learned coefficients (from largest to the smallest) in terms of the absolute value, and then we select the top $53 \times N$ coefficients for each N th gene in the x-axis. We detail this in the manuscript (SI Page 8, Line 34).

Here are the N s that correspond to the total number of nonzero coefficients of ElasticNet in each of the consistency experiments:

Training using 12 samples in batch 1 (first fold for Figure 2A): 4,829

Training using 12 samples in batch 2 (second fold for Figure 2A): 12,419

Training using 30 patient samples (Figure 2B): 8,946

At each of these points in the x-axis, all nonzero N s are used up. After that, the gene-drug pairs are not ordered based on the coefficient strength, but randomly, because all the remaining coefficients are zero. Therefore, the poor performance of ElasticNet is unlikely to be caused by the zero coefficients, because the ElasticNet curves become almost flat after $N = 2,000$ in Figure 2A and after $N=6,000$ in Figure 2B. The poor performance is very likely because the strong gene-drug associations tend to be not conserved in the validation data, potentially because of the three reasons described above.

Based on our responses, if the reviewer is still concerned about the ElasticNet curves in Figure 2, we will remove the curves that correspond to ElasticNet and multi-task learning method from Figure 2 in the final round.

[8] A minor point: Figure 5A appears to visualize the \log_{10} p-values rather than the $-\log_{10}$ p-values.

We thank the reviewer for pointing it out. We updated colorbar in Figure 5A as the reviewer suggested in the revised manuscript.

[9] A minor point: Figure 4A could be inset into the white space of Figure 4B to conserve space.

We thank the reviewer for the suggestion. We updated Figure 4 in the revised manuscript as the reviewer suggested.

[10] At the minimum, the Supplementary Materials should include Figures such as 5A and 5B but corresponding to genes selected under Pearson's Linear Correlation (and Spearman's as well, possibly) and the ElasticNet. Does MERGE truly provide a path towards identifying the roles (as detailed on pages 8 and 9) of FLT3, CASP8AP2, L2HGDH, MNT, BAZ2B, MZF1, BEX2, and SMARCA4 that other methods would not have?

We thank the reviewer for the excellent suggestion. We presented the heatmaps for ElasticNet, multi-task learning, Pearson's P-value, and Spearman P-value methods in Figure S4 in the Supplemental Information. In response to the reviewer's comment, we also added Table S7 to the Supplemental Information, which shows whether or not each of FLT3, CASP8AP2, L2HGDH, MNT, BAZ2B, MZF1, BEX2, and SMARCA4 is identified by the four methods we compared with MERGE in Figure 2 and 3C.

Except for ElasticNet, all the other 3 methods identified only 1 of these 8 genes, L2HGDH. ElasticNet identified three of these 8 genes, CASP8AP2, L2HGDH, and BEX2. Five of these 8 genes (SMARCA4, FLT3, MNT, BAZ2B and MZF1) would not have been identified by any of these 4 methods. These results show that the majority of these 8 genes would not have been identified by alternative methods. We incorporated these results into the revised manuscript (Page 8 Line 17).

[11] On line 6 of page 3 of Suppl. Materials, the authors state that the driver features were standardized. How were the "Candidate regulator" and CNV scores standardized? Are those driver features purely binary?

As we stated in the Supplemental Experimental Procedure 4, the candidate regulator and CNV scores are purely binary. They are standardized (i.e., centered and scaled to have mean 0 and standard deviation 1) similarly to other non-binary features. Perhaps, the reviewer was wondering whether it makes sense to standardize a binary variable. For a statistical method that combines data from several different kinds of variables (i.e. features) and aims to learn weights for the variables with regularization, it is required to provide all variables to the algorithm on the same scale, for the sake of fairness to all variables. Binary variables are not exceptions. For example, Rob Tibshirani said that standardization is required for dummy (binary) variables when using LASSO: "For categorical regressors, one codes the regressor with dummy variables and then standardizes the dummy variables." [1]. This statement can be extended to MERGE given that MERGE is also a regularized regression approach incorporating thousands of variables (i.e. gene expression levels) as well as multiple kinds of driver features (i.e. MERGE features) into a single statistical framework, and learning the importance of each variable as well as each driver feature. We added this discussion to the revised manuscript (SI Page 3 Line 27).

[1] Tibshirani R. The LASSO method for variable selection in the Cox model. Statistics in Medicine Vol. 16, 385-395 (1997).

[12] The discussion section would be improved by including some comments concerning the Demographic and clinical attributes of the 30 AML patients from which the sample materials were obtained. What larger, more general clinical population, can the results be postulated as being relevant to?

We thank the reviewer for the very useful comment. We provide all clinical data in terms of usual evaluation, response to treatment and duration of remission in Table S1. We also have Supplemental Experimental Procedure 2 in the revised manuscript, where we make a short summary of the clinical information of the patients (SI Page 1, Line 23). The 30 AML patients were consecutively enrolled on a protocol to obtain laboratory samples for research. They were selected solely on the basis of sufficient leukemia cell numbers. There are 24 newly diagnosed and 6 relapsed (1st or 2nd relapse) patient samples. The majority of the samples (19 out of 30) are males. Median age is 54 and the age range is [19, 83]. Six patients had antecedent hematologic disorder. Seven had an NPM1 mutation and 7 had a FLT3 mutation. According to European LeukemiaNet criteria, 7 samples are in favorable risk group, 11 are in intermediate-1, 2 are in intermediate-2, and 10 are in the adverse risk cytogenetics group. As the samples were consecutively obtained and not selected for any particular attribute, they are representative of patients seen at a tertiary referral center. We incorporated this information into the revised manuscript (Page 12, Line 5).

Does the inclusion of “prior” information from cell lines and other sources, suggest that it is possible that the results could be considered more generalizable than those obtained by an analysis only incorporating the data from the 30 AML samples?

First of all, we want to point out that we do not include any prior information from the cell lines in the current version of the MERGE approach. All sources from which we use prior information to train the MERGE model are from primary patient samples. We used cell lines only for validation of the identified gene-drug associations (Figure 2B). However, MERGE is a general framework and it would be a straightforward extension to include prior information from cell lines as well.

Since each of the datasets that we collect prior information (driver features) from contains a much higher number of samples than 30 (e.g. TCGA AML data from nearly 200 patients), it is highly likely that the results of MERGE are more generalizable to larger clinical populations than the methods that incorporate the data from 30 AML samples. In fact, Figure 2 exemplifies this aspect of MERGE. We incorporated this discussion into the revised manuscript (Page 12, Line 9).

[13] An experimental design involving a truly independent validation arm (e.g., involving an additional 30 AML patients) would have dramatically improved the potential impact of the work.

We thank the reviewer for the suggestion. In response to the reviewer’s comment, we generated the drug response and the expression data from 12 additional patient samples. Using these data as validation data, we measured the consistency of the significant gene-drug associations learned from the initial 30 samples on these new validation data from additional 12 samples. As presented in Figure 2C, MERGE outperforms 4 alternative approaches. We added these results to the revised manuscript (Page 5 Line 35). Unlike the initial 30 samples, the new 12 samples are refractory AML patients, which might be a reason for a relatively lower consistency rate of MERGE in Figure 2C compared to the split sample test (Figure 2A). We do not expect that highly refractory patients (refractory after 3-6 prior regimens) will exhibit the same drug sensitivity patterns as the 30 patients with primarily newly diagnosed disease or 1st or 2nd relapse. In addition, these refractory patients may have activated other survival pathways and mechanisms of drug resistance. Still, even in this challenging validation setting, MERGE outperforms 4 alternative methods (Figure 2C).

[14] By what method was the FDR estimated in the various components of the analysis?

We estimated FDR based on the method introduced by [1]. We added the reference to the manuscript (SI Page 4 Line 23). We thank the reviewer for pointing it out!

[1] Benjamini Y & Hochberg Y. *Controlling the false discovery rate: a practical and powerful approach to multiple testing*. *Journal of the Royal Statistical Society Series B* 57, 289–300 (1995).

[15] *Did authors consider summary methylation and CNV measures other than sample mean?*

We thank the reviewer for the comment. First of all, we want to clarify that we, in fact, did not use sample mean for computing the CNV score of each gene. As we described in the Supplemental Experimental Procedure 4, we set the CNV score of a gene to 1 if the gene is amplified or deleted by at least .05 in at least 20 out of 191 patients (~10%), and to 0 otherwise.

As the reviewer pointed, for methylation, we used sample mean across TCGA/AML patients to compute the methylation scores. While there may be other ways to get a summary statistic across all patients, we considered only sample mean in the current study, given that mean is the most commonly used measure of central tendency for continuous-valued variables, and several authors have averaged methylation values over samples (e.g. [1]). We included this discussion in the Supplemental Experimental Procedure 5 in the revised manuscript (SI Page 3 Line 16).

[1] Moarii M, Boeva V, Vert JP & Reyal F. *Changes in correlation between promoter methylation and gene expression in cancer*. *BMC Genomics* 16:873 (2015).

For methylation profiles, in particular, was there any concern that averaging across all CpG results (mapped to a given gene) might be less efficient than a feature selected (and hopefully more informative than the average) subset of CpGs?

We thank the reviewer for the great point. We could have designed the MERGE algorithm such that it automatically selects a subset of CpGs used to compute the MERGE score. However, that will greatly increase the number of parameters (i.e., number of CpG sites) and add one additional sparsity tuning parameter for these parameters. This challenge will require additional computational innovation. The main contribution of our paper is a novel computational approach to integrate prior knowledge on genes' potential to drive diseases into the biomarker discovery problem. Further developments, such as selecting a subset of CpG sites or incorporating the functional annotation (e.g., promoter, etc) of the CpGs' regions, can be great follow-up projects but would be out of the scope of the current paper. Also, given that we observed a strongly negative v_k value for the methylation feature (Figure 4A), which is well consistent with our prior knowledge that methylation in a promoter region silences the corresponding gene (Page 7, Line 41), we believe that the gene-based methylation data we used are informative of biomarker potential.

We describe below how we chose the gene-based methylation profile out of the 4 profiles provided by TCGA. TCGA AML study has preprocessed methylation data generated by 4 different pipelines that are used to retrieve the gene-based methylation values from the CpG probes:

- 1) using the CpG probe that has the highest anticorrelation with the gene's *expression* level,
- 2) using the CpG probe that has the highest anticorrelation with the *clinical* data,
- 3) using the mean signal intensities across all CpG probes in the proximity of the gene, and

4) using the CpG probe that has the maximum standard deviation across all beta values.

We agree that a specific position of methylation (e.g. whether it occurs around the transcription start site or over a gene body) is important information to understand the epigenetic cause of variation of downstream phenotypes such as gene expression levels. Accordingly, the data generated by the approaches #1) and #2) above are generally chosen when the goal is to explain a phenotype of interest (i.e. gene expression or clinical data) and choose the probe likely to represent the molecular basis for the phenotype. Since our goal is to develop a general framework to utilize methylation data from the external AML sources, in our study, we used the data generated by the approach #3), which is independent from any specific phenotype and gives a general summary of the TCGA methylation data. We did not use the data from 4) because it discards any probes with a standard deviation below a specified cutoff, so the dataset contains only ~2K genes which is too small compared to the ~17K genes in the MERGE model.

We incorporated this information into the Supplemental Experimental Procedure 5 in the revised manuscript (SI Page 3 Line 2).

Reviewer 2

This is an interesting and technically strong paper to identify gene expression markers of drug response in acute myeloid leukemia for many classes of therapeutics. The study contributes both a valuable data set – namely gene expression data for 30 AML patient-derived specimens, together with dose response curves in these samples measured for 160 drugs – and a simple but principled approach to incorporate prior knowledge about the gene relevance into the computational model for identifying gene-drug associations, which is called the MERGE algorithm. The MERGE optimization problem modifies a standard regression approach for predicting drug responses from gene expression values by simultaneously learning a common weighting over 5 gene features (mutations in TCGA/AML, expression “hubness”, “regulator” status, CNVs from TCGA/AML, methylation from TCGA/AML); the weighted features or MERGE score appears in a weighted L2 regularization function in the joint learning of regression models for all the drugs, enforcing a shared set of prior on relevant drugs. The authors show that MERGE improves consistency of gene-drug associations between training and test sets compared to standard elastic net and identifies known or plausible associations for several classes of drugs. They also provide experimental validation for a novel association between expression of SMARC4 and response to topoisomerase II inhibitors by showing that overexpression of SMARC4 in a leukemia cell lines with low/medium endogenous expression tends to sensitize cells to these drugs (though some of these cell lines are already sensitive to specific topoisomerase II inhibitors without SMARC4 overexpression).

Overall this is a solid paper that provides some evidence that the method can find novel gene-drug associations that may be worth exploring further. The main weakness is that further method comparison and statistical assessment is needed, since there has been much recent work on predicting drug response from molecular data in cell lines. Specific additional analyses are suggested below:

(1) The method is evaluated based on consistency of gene-drug associations between training and test sets and compared to elastic net and a simple correlation statistic by this measure. However, the MERGE algorithm actually solves a regression problem, and most work in this area trains predictive models and assesses regression performance on the test set. Here the data sets are all small (30 new patient-derived samples, 12 cell lines in CCLE), but it would still be helpful to try to assess whether

MERGE outperforms elastic net as a regression model on held out cell lines, or is the advantage only for expression marker identification?

We thank the reviewer for the great point. In response to the reviewer's suggestion, we performed additional experiments to assess the prediction performance of MERGE. We added two more methods which reviewer recommended below in comment #2 below. We compared MERGE to 3 other methods (ElasticNet, Multi-task learning [1], and Bayesian multi-task multiple kernel learning method that won the NCI-DREAM Drug Sensitivity Prediction Challenge [2]) in 2 different settings: 1) Leave-one-out cross-validation (LOOCV) on 30 patient samples, and 2) Split sample test (where we train the models using 12 samples in batch 1 and test using the 12 samples in batch 2, vice versa).

We compared among these methods based on the evaluation metric used in the NCI-DREAM challenge -- accurately predicting the ranking of patients' response to each drug, measured by comparing between the predicted (measured via LOOCV) and actual drug sensitivity. As shown in the Figure 7 in the revised manuscript, in both of the experimental settings, MERGE outperforms all other methods in terms of the prediction performance averaged over all drugs. Moreover, as shown in Figure S6, MERGE achieves a better prediction performance for the majority of the drugs (30 out of 53). We revised the manuscript such that we include the results of the prediction experiments (Page 5, Line 8; Page 7, Line 5).

[1] Pong TK, Tseng P, Ji S & Ye J. Trace Norm Regularization: Reformulations, Algorithms, and Multi-Task Learning. *SIAM Journal on Optimization*, 20(6): 3465–3489 (2010).

[2] Costello JC et al. A community effort to assess and improve drug sensitivity prediction algorithms. *Nature Biotechnology*, Dec;32(12):1202-12 (2014).

(2) There are a number of other recently published methods for learning to predict drug sensitivity, and it would be informative to compare these methods to MERGE in terms of both prediction accuracy and consistency of gene-drug associations. Of particular interest are multi-task methods, since the MERGE score-based L2 regularizer shares information across the drug tasks. It would be straightforward to compare to the recently published trace norm multi-task drug prediction approach (Yuan, Paskov et al, Nature Sci Reports 2016). Another option is the Bayesian multitask kernel method that was the winner in a recent DREAM competition (Costello et al., Nat Biotechnol 2014), though as a kernel method, this approach gives prediction performance more readily than gene-drug associations.

We appreciate the reviewer's excellent suggestion. As suggested, in the revised manuscript we included both of Bayesian multi-task multiple kernel learning (MKL) (the winner of the DREAM competition - Costello et al., *Nat Biotechnol* 2014) and the multi-task learning (Yuan, Paskov et al, *Nature Sci Reports* 2016) whose implementation was available in [1]. We included multi-task learning to both of our experiments for testing the prediction performance (Figure 7) and the consistency of gene-drug associations (Figure 2; Figure 3C). As the reviewer mentioned, the DREAM winning method is a kernel method, and so it does not learn a weight for each feature (here, gene). In that case, it is not clear how to order the gene-drug associations according to this method and check the associated consistency or drug class specificity; so we included MKL to our prediction experiments only (Figure 7). MERGE outperforms all methods including these two methods; consistency of gene-drug associations (Figure 2), drug class specificity of the genes (Figure 3C) and predicting a ranked list of patients' drug responses (Figure 7). We added these new results to the revised manuscript (Page 5, Line 26; Page 7, Line 7).

[1] Pong TK, Tseng P, Ji S & Ye J. Trace Norm Regularization: Reformulations, Algorithms, and Multi-Task Learning. *SIAM Journal on Optimization*, 20(6): 3465–3489 (2010).

(3) FLT3 mutation status in individual cell lines is not used in the MERGE model, although patient mutation status was collected. For many reasons both practical and historical, patient mutation status is far more likely than expression level to be measured and used for making therapeutic decisions. The authors state that their finding of an association between FLT3 gene expression level and drug response for multiple agents (including FLT3 inhibitors) is novel. It would be valuable to explore this issue more deeply – at least to quantify whether the gene expression level gives a more statistically significant marker of response than mutation status, at least in the drug response assays.

We thank the reviewer for the great point. We agree that the FLT3 mutation status is an important prognostic indicator in AML with the potential to guide therapy. In response to the reviewer’s comment, we compared between FLT3 mRNA expression and mutation status in terms of the statistical significance of association with drug response of the patients to the 53 drugs that we consider in our study. As shown in the Figure S5 in the revised manuscript, both for (A) patients and (B) cell lines, mRNA level achieves a more significant association than the FLT3 mutation status for a much larger number of drugs: (A) 36 drugs (expression) vs. 17 drugs (mutation); and (B) 31 drugs (expression) vs. 22 drugs (mutation) in cell lines. We used [1] as the source for the FLT3 mutation status for the cell lines. We described this result in the revised manuscript (Page 8 Line 41).

[1] Quentmeier H, Reinhardt J, Zaborski M & Drexler HG. FLT3 mutations in acute myeloid leukemia cell lines. *Leukemia*, 17(1): 120–4 (2003).

Minor issues:

- Figure 7 seems like it could be concatenated with Figure 6 or moved to the supplement – the panels are just confirming expression/overexpression of SMARC4 and not describing drug response validations

We thank the reviewer for the suggestion. The Figure 7 in the original manuscript is now concatenated with Figure 6 in the revised manuscript. We have a new Figure 7 that shows the comparison of the prediction accuracy results between MERGE and 3 other methods.

- “the repertoire of drugs for cancer is rapidly expanding” – this statement in the introduction is not generally considered to be true; rather, it is expensive to develop a new drug and easier to repurpose an existing agent

We thank the reviewer for the helpful comment. We agree with the reviewer that the expansion is not rapid. In response to the reviewer’s comment, we removed the word “rapidly”, replaced “drugs” with “potential drugs” (Page 1, Line 19). Also, in the main text, we mentioned (with citations) the number of drugs that are currently in clinical development in the U.S. (Page 1, Line 41).

- “which is a more challenging setting and the testability” – non-grammatical statement, and in general the “testability” language is confusing

- DNase rather than DNase

We thank the reviewer for the comments. Those are now corrected.

Reviewer 3

Manuscript by Lee et al presents data from gene expression and drug screening of 30 AML patients and 14 AML cell lines. Authors propose use of MERGE a computational method that uses genes biological importance/function and identified gene expression signatures for drug response. Authors propose that using multi-dimensional information on a gene helps in determine its potential as a cancer driver which can impact its influence on drug sensitivity. Overall the study is novel and nicely designed however there are some major concerns as listed below:

1. Results: Author indicate that 12 of 15 drugs that are used in clinic for AML treatment were associated with CR, one of the most commonly used drug cytarabine is missing from this list, authors should provide some comments on this.

We thank the reviewer for the great point. The 15 drugs listed in Table S2 as drugs used in clinic for AML treatment are a subset of the 53 drugs included in the MERGE analysis. While cytarabine is commonly used to treat AML and the majority of the 30 AML patients received cytarabine, it was not one of the 53 drugs included in our computational framework. In this initial study of MERGE method development, to increase the statistical power of the MERGE learning, we focused on the drugs whose responses significantly vary across patients. Our criteria for inclusion of drugs is to exhibit activity (cell viability \leq 50%) of the drug against at least half of the 30 patient samples, and cytarabine did not satisfy this criterion. One possible reason why cytarabine did not satisfy our criterion is that, the tested concentrations were too low for the 12 patients in the 1st set tested (concentration range 2e-10 M to 1e-6 M), and cytarabine did not exhibit any activity for those patients until the highest concentration point. The concentrations were then increased for the next 18 patient samples (range 4e-8 M to 1e-4 M) For comparison, the peak mean plasma concentration after high dose cytarabine is 2e-3 M [1]. We incorporated this information into the revised manuscript (Page 4, Line 9; SI Page 9, Line 17).

[1] Hande KR, Stein RS, McDonough DA, Greco FA & Wolff SN. Effects of high-dose cytarabine. *Clinical Pharmacology and Therapeutics*, 31(5): 669–74 (1982).

Authors should provide information on of the drugs with significant association between drug sensitivity and complete remission how many agents were actually used to treat the cohort of 30 patients.

We appreciate the reviewer's suggestion. Several different regimens were used to treat the 30 patients, including eight receiving 7+3 (cytarabine + anthracycline) with either daunorubicin or idarubicin, with one patient receiving sorafenib, one azacitidine, and 2 gemtuzumab as 3rd agents, three flavopiridol/cytarabine/mitoxantrone, six clofarabine and cytarabine with or without G-CSF, two received fludarabine and cytarabine with G-CSF (one with gemtuzumab, one with idarubicin), one mitoxantrone/etoposide/cytarabine with investigational CXCR4 inhibitor, and one investigational drug with cytarabine, one GMCSF/azacitidine/gemtuzumab, one decitabine, one was not treated, one cytarabine/etoposide/gemtuzumab/cyclosporine. Because so many different drug regimens were used to treat this relatively small number of patients, it was not possible to perform correlation analysis between

the clinical regimen and the drug testing. We listed the agents we used for treatment in the revised manuscript (SI Page 1, Line 28).

2. Although authors point out that MERGE algorithm uses relative importance of each driver feature details on this are lacking and should be provided. Similarly criteria for driver feature characterization should be provided.

We thank the reviewer for the suggestion to provide more details on:

- 1) *How MERGE uses relative importance of each driver feature:* We assume that the reviewer is referring to the sentence, “The driver feature weights learned by MERGE algorithm indicate the relative amount of importance of each driver feature on the MERGE score” (Page 7, Line 24). We realized that in the original manuscript, the details about the formulation did not appear in the Results section but in the Supplemental Information (SI Page 5, Line 3). In response to the reviewer’s suggestion, we briefly described how the MERGE score is computed based on the driver feature weights and driver features (Page 7, Line 25) and referred to the Supplemental Experimental Procedure 10 where we described the probabilistic model of MERGE in detail.
- 2) *Criteria for driver feature characterization:* We indeed provided the details on how the driver features were extracted are in the Supplemental Experimental Procedure 4 of the manuscript (SI Page 2, Line 11). In the revised manuscript, we clearly referred to the Supplemental Experimental Procedure 4 in the Results section (Page 4, Line 31; Page 7, Line 25).

We thank the reviewer for encouraging us to improve the manuscript by making this detailed information more easily found in the revised manuscript.

3. Authors should also provide information of risk group category/ cytogenetic features for these 30 patients and whether or not these were taken into account in the association analysis.

We thank the reviewer for the suggestion. As referred in the second paragraph of the Results section (Page 3, Line 38), Table S1 contains the information of the risk group category/cytogenetic features for each patient. We also summarized risk group information in Supplemental Experimental Procedure 2 in the revised manuscript (SI Page 1, Line 26).

We did not take risk group category/cytogenetic features into account in the association analysis in our MERGE framework. This is because our main goal is to build a general tool that addresses the high-dimensionality challenge (i.e., number of samples \ll number of genes) and make an efficient use of the gene expression data and prior knowledge and data, in order to identify true positive associations.

In response to the reviewer’s concern, to consolidate our finding, we performed a covariate analysis to confirm that the top-ranked gene-drug associations discovered by MERGE are still conserved when the risk group/cytogenetic features are taken into account. For this purpose, we checked whether the associations of the genes shown in the heatmap in Figure 5B with the corresponding drugs (highlighted as red or green) are still conserved when we add each of the followings as an additional covariate to the linear model: 1) Cytogenetic risk, 2) FLT3 mutation status, 3) NPM1 mutation status. We show the result of this analysis in Figure S8, where each dot corresponds to a gene-drug pair, and each color to a different covariate. Most of the dots being closer to the diagonal implies that the significance of associations does not decrease much after adding the covariates. Moreover, out of 357 dots, only 8 of them are below the horizontal red line, which means that 98% of the gene-drug associations found by MERGE are still

significant ($p \leq 0.05$) even after taking into account the covariate. We added this result to the Discussion section in the revised manuscript (Page 11, Line 39).

Reviewers' comments:

Reviewer #2 (Remarks to the Author):

Overall, the authors have done a good job in adding new method comparison results to address comments from the previous review. Instead of simply comparing feature consistency of their MERGE algorithm against “vanilla” elastic net, they now compare against two more state-of-the-art methods, trace norm multitask regression (Yuan, Paskov et al.) and a Bayesian multiple kernel learning (MKL) approach that won a recent DREAM challenge (Costello et al.), for feature consistency (in the trace norm case) and prediction accuracy (for both multitask methods). The method comparison mostly seems solid but the new added text tends to misrepresent/overstate the extent of and reasons for the performance differences. Therefore, to provide a more balanced assessment of the advantages of their MERGE algorithm, I recommend a few additional analyses, which should be quick to do:

(1) The authors claim that MERGE outperforms trace norm multitask learning and MKL, but the overall performance difference reported for example in Figure 7 does not look statistically significant: mean Spearman correlation of .46 or .47 versus .48 for MERGE on held-out test sets, with only a slightly larger difference in the LOOCV setting. Are these performance differences statistically significant by a Wilcoxon signed rank test? — the p value should be reported. Even if there is mild statistical significance, there seems to be no practical difference in the reported Spearman correlations for MERGE vs. the multitask methods, which line up along the diagonal in the scatter plot. It might be better to just state that MERGE is competitive with recent multitask methods rather than claiming a meaningful win in prediction accuracy.

(2) The authors do show much more feature consistency of MERGE compared to elastic net and trace norm multitask learning. However, some of these results are highly puzzling and perhaps are presented/interpreted in the wrong way. E.g. in Figure 2A, simple correlation measures outperform all methods for feature consistency for the very top-ranked genes (which is worth mentioning in the text), while multitask learning is doing somehow worse than elastic net at choosing consistent features between models trained on two different sets of 12 patients. However, as the feature consistency task gets harder — i.e. the models are trained on two more disparate data sets, like patients vs. cell lines — the performance of multitask learning starts improving relative to elastic net. Indeed, normally we would expect that multitask learning would always improve feature consistency over elastic net since the individual task models become more similar to each other. My interpretation of these results is that the problem in Figure 2A is elastic net and trace norm multitask regression are being trained on too few training examples ($n=12$). A little more caution in presenting these results is needed.

(3) Also on the feature consistency analysis, results are shown for top N ranked genes for N going up to 14K or 16K genes, but obviously the most relevant results are for small N, maybe only the top 500 genes. Meanwhile, the strong consistency of MERGE even in settings where there is perhaps not enough data to accurately train an elastic net model is likely due to the strong prior. The prior may also be the reason why there is slightly higher than expected feature consistency even among the top 10K genes. To understand this result better, the authors could perform the following experiment: randomize the training data — e.g. (1) randomize each gene value over training examples or (2) randomize the response variable, so that it is not possible to learn meaningful prediction models — and then report the MERGE feature consistency in the settings A, B, and C of Figure 2. This will show how much the prior is giving feature consistency even in the absence of meaningful learning from the training data. This can be reported as a supplementary figure — it is just to provide caution about over-interpreting the consistency results.

Overall, the method provides a nice way of incorporating prior information in a small data setting, and the paper is almost ready for publication. The authors just need to clear up the issues above with comparison to methods that do not use feature priors. E.g. if in Figure 2A, none of the machine learning methods are learning anything, and the consistency of MERGE is due to the prior alone, than perhaps it is better to do a different analysis (consistency on larger but overlapping subsets of patients?).

Minor:

- "better prediction performance for majority of the drugs" -> "...for a majority..."

Reviewer #3 (Remarks to the Author):

Authors have addressed the comments adequately

Reviewer #4 (Remarks to the Author):

My task was to review the authors' responses to Reviewer 1's remarks, as Reviewer 1 is unavailable to check these responses.

I can state that every point raised by Reviewer 1 has been addressed in a convincing manner, with solid argumentation. I have checked item by item the corresponding modified sections in the manuscript, as well as in the supplementary files. Besides, my reading of this excellent paper raised no novel point that would need further revision. Therefore, my advice is that this paper can be accepted for publication.

We are grateful to reviewers for their careful consideration of this manuscript and their useful suggestions for improvement. In response to Reviewer 2's comments, we made changes to improve the manuscript and address the reviewer's concerns.

Reviewer 2

Overall, the authors have done a good job in adding new method comparison results to address comments from the previous review. Instead of simply comparing feature consistency of their MERGE algorithm against "vanilla" elastic net, they now compare against two more state-of-the-art methods, trace norm multitask regression (Yuan, Paskov et al.) and a Bayesian multiple kernel learning (MKL) approach that won a recent DREAM challenge (Costellow et al.), for feature consistency (in the trace norm case) and prediction accuracy (for both multitask methods). The method comparison mostly seems solid but the new added text tends to misrepresent/overstate the extent of and reasons for the performance differences. Therefore, to provide a more balanced assessment of the advantages of their MERGE algorithm, I recommend a few additional analyses, which should be quick to do:

(1) The authors claim that MERGE outperforms trace norm multitask learning and MKL, but the overall performance difference reported for example in Figure 7 does not look statistically significant: mean Spearman correlation of .46 or .47 versus .48 for MERGE on held-out test sets, with only a slightly larger difference in the LOOCV setting. Are these performance differences statistically significant by a Wilcoxon signed rank test? — the p value should be reported. Even if there is mild statistical significance, there seems to be no practical difference in the reported Spearman correlations for MERGE vs. the multitask methods, which line up along the diagonal in the scatter plot. It might be

better to just state that MERGE is competitive with recent multitask methods rather than claiming a meaningful win in prediction accuracy.

As the reviewer recommends, we computed the p -values for the Wilcoxon signed rank test for each comparison between MERGE and an alternative method in Figure 7A-B. We report those p -values below:

Held-out test sets (Figure 7A):

ElasticNet vs. MERGE - $p=0.037$

DREAM vs. MERGE - $p=0.1$

Multitask vs. MERGE - $p=4.1e-3$

LOOCV (Figure 7B):

ElasticNet vs. MERGE - $p=2e-3$

DREAM vs. MERGE - $p=5.7e-5$

Multitask vs. MERGE - $p=6.1e-3$

The p -values are significant at a $p \leq 0.007$ level for four out of six comparisons, and, at a $p \leq 0.1$ level for the other two. We showed these p -values in the legend of Figure 7A-B (Page 26 Line 9) and added text on these significance measures to the manuscript (Page 7, Line 24). We replaced “outperforms all alternative methods” with “performs competitively with the alternative methods” (Page 7, Line 23), as the reviewer suggests.

We note that while working on the response to this review, we discovered an error in the code used to generate Figure 7, which led to a slightly lower ElasticNet performance. We corrected this error in this revision.

(2) The authors do show much more feature consistency of MERGE compared to elastic net and trace norm multitask learning. However, some of these results are highly puzzling and perhaps are presented/interpreted in the wrong way. E.g. in Figure 2A, simple correlation measures outperform all methods for feature consistency for the very top-ranked genes (which is worth mentioning in the text), while multitask learning is doing somehow worse than elastic net at choosing consistent features between models trained on two different sets of 12 patients. However, as the feature consistency task gets harder — i.e. the models are trained on two more disparate data sets, like patients vs. cell lines — the performance of multitask learning starts improving relative to elastic net. Indeed, normally we would expect that multitask learning would always improve feature consistency over elastic net since the individual task models become more similar to each other. My interpretation of these results is that the problem in Figure 2A is elastic net and trace norm multitask regression are being trained on too few training examples ($n=12$). A little more caution in presenting these results is needed.

We appreciate the careful reading and assistance in correcting an error in Figure 2A. We realized that in Figure 2A we had not applied genome-wide FDR correction on the Pearson’s correlation p -values of the gene-drug associations for the individual batches, while we did apply genome-wide FDR correction for settings with 30 samples (previous Figure 2B-C), as we mentioned in the Supplementary Material (SI Page 4, Line 18). After we corrected this error and applied genome-wide FDR correction to the Pearson’s p -values from the 12 samples in each of batch 1 and batch 2, we observed that batch 1 is left with zero significant p -values (at the FDR-corrected p -value threshold .1), which would lead to zero consistency for Figure 2A for any number of top-ranked genes (N).

We believe that the lack of significant correlations after FDR correction when 12 samples are used indicates that $n=12$ is too low a sample size. We agree with the reviewer that performing experiments on 12 samples for training and testing would not lead to a reliable result. Therefore, in the revised manuscript, we excluded the experiments for which we used 12 samples for training and testing (Figure 2A, and Figure S9). As shown in Figure 2 in the revised manuscript, simple correlation measures generally underperform most of the methods, and multitask learning outperforms all methods except MERGE, as the reviewer correctly expected.

(3) Also on the feature consistency analysis, results are shown for top N ranked genes for N going up to 14K or 16K genes, but obviously the most relevant results are for small N, maybe only the top 500 genes. Meanwhile, the strong consistency of MERGE even in settings where there is perhaps not enough data to accurately train an elastic net model is likely due to the strong prior. The prior may also be the reason why there is slightly higher than expected feature consistency even among the top 10K genes. To understand this result better, the authors could perform the following experiment: randomize the training data – e.g. (1) randomize each gene value over training examples or (2) randomize the response variable, so that it is not possible to learn meaningful prediction models – and then report the MERGE feature consistency in the settings A, B, and C of Figure 2. This will show how much the prior is giving feature consistency even in the absence of meaningful learning from the training data. This can be reported as a supplementary figure – it is just to provide caution about over-interpreting the consistency results.

We agree with the reviewer that the use of the strong prior could be the reason why MERGE performed well. To show that the training data (i.e., expression and drug sensitivity data from 30 patients) were also a key factor and MERGE can learn from these data, we followed the reviewer's suggestion to carry out an additional experiment. We performed MERGE on 100 different data permutations where training samples in the drug response data were shuffled (by doing (2), as the reviewer suggested). We then tested the learned models on both 14 cell lines and the 12 refractory samples (Figure 2). We included this result as a supplementary figure (Figure S9). As can be seen in Figure S9, the MERGE run using the true sample ordering achieved a higher consistency rate than most of the permutations, and many of the permutations actually performed worse than the competing methods. This shows that training data can never be dismissed, and the prior information itself is not very helpful when training data are random. We added this discussion to the manuscript (Page 6, Line 15).

Overall, the method provides a nice way of incorporating prior information in a small data setting, and the paper is almost ready for publication. The authors just need to clear up the issues above with comparison to methods that do not use feature priors. E.g. if in Figure 2A, none of the machine learning methods are learning anything, and the consistency of MERGE is due to the prior alone, than perhaps it is better to do a different analysis (consistency on larger but overlapping subsets of patients?).

We appreciate the reviewer's positive input. We believe that the additional experiments we performed and summarized above will help convince the reviewer that the high consistency of MERGE is not due to the prior alone, but that MERGE in fact learns critical information from the training data, which significantly contributes to its high consistency.

Minor:

- "better prediction performance for majority of the drugs" -> "...for a majority..."

We appreciate your noting this omission and corrected the error in the revised manuscript.

We appreciate this considerable time and effort to review.

REVIEWERS' COMMENTS:

Reviewer #2 (Remarks to the Author):

The authors have addressed my previous critiques in a satisfactory manner.